# Recognition and cleavage of human tRNA methyltransferase TRMT1 by the SARS-CoV-2 main protease

Angel D'Oliviera[1], Xuhang Dai[2], Saba Mottaghinia[3], Sophie Olson[1], Evan P Geissler[1], Lucie Etienne[3], Yingkai Zhang[2,4], Jeffrey S Mugridge[1]*

[1]Department of Chemistry & Biochemistry, University of Delaware, Newark, United States; [2]Department of Chemistry, New York University, New York, United States; [3]CIRI (Centre International de Recherche en Infectiologie), Univ Lyon, Inserm, U1111, Université Claude Bernard Lyon 1, CNRS, UMR5308, ENS de Lyon, Lyon, France; [4]Simons Center for Computational Physical Chemistry at New York University, New York, United States

## eLife Assessment

This manuscript provides **important** structural insights into the recognition and degradation of the host tRNA methyltransferase TRMT1 by SARS-CoV-2 protease nsp5 (Mpro). The data provide **compelling** support for the main conclusions of the authors. These results will be of interest to researchers studying structures, substrate recognition and specificity of viral proteases and their action on cellular targets.

*For correspondence:
mugridge@udel.edu

Competing interest: The authors declare that no competing interests exist.

**Abstract** The SARS-CoV-2 main protease (M$^{pro}$ or Nsp5) is critical for production of viral proteins during infection and, like many viral proteases, also targets host proteins to subvert their cellular functions. Here, we show that the human tRNA methyltransferase TRMT1 is recognized and cleaved by SARS-CoV-2 M$^{pro}$. TRMT1 installs the $N^2,N^2$-dimethylguanosine (m2,2G) modification on mammalian tRNAs, which promotes cellular protein synthesis and redox homeostasis. We find that M$^{pro}$ can cleave endogenous TRMT1 in human cell lysate, resulting in removal of the TRMT1 zinc finger domain. Evolutionary analysis shows the TRMT1 cleavage site is highly conserved in mammals, except in Muroidea, where TRMT1 is likely resistant to cleavage. TRMT1 proteolysis results in reduced tRNA binding and elimination of tRNA methyltransferase activity. We also determined the structure of an M$^{pro}$-TRMT1 peptide complex that shows how TRMT1 engages the M$^{pro}$ active site in an uncommon substrate binding conformation. Finally, enzymology and molecular dynamics simulations indicate that kinetic discrimination occurs during a later step of M$^{pro}$-mediated proteolysis following substrate binding. Together, these data provide new insights into substrate recognition by SARS-CoV-2 M$^{pro}$ that could help guide future antiviral therapeutic development and show how proteolysis of TRMT1 during SARS-CoV-2 infection impairs both TRMT1 tRNA binding and tRNA modification activity to disrupt host translation and potentially impact COVID-19 pathogenesis or phenotypes.

## Introduction

SARS-CoV-2 has resulted in over 6 million deaths worldwide since the start of the COVID-19 pandemic in early 2020 (**G. W. H. Organization, 2023**; **C. for S. Science, E, 2023**). The development of mRNA and other vaccines has played a large and critical role in reducing mortality since their introduction

**Figure 1.** A peptide sequence found in human TRMT1 fits the cleavage consensus for the SARS-CoV-2 main protease (M$^{pro}$). (**A**) Overview of the structure of the SARS-CoV-2 M$^{pro}$ homodimer (PDB 7BB2) with substrate peptide residues (P4-P3-P2-P1-P1'-P2'-P3'-P4') illustrated in the M$^{pro}$ active site (inset); proteolytic cleavage takes place between substrate residues P1 and P1' (dotted line). (**B**) The TRMT1(527–534) sequence found in a linker region between the TRMT1 SAM methyltransferase (MTase) and zinc finger (ZF) domains is consistent with the SARS-CoV-2 M$^{pro}$ cleavage consensus sequence.

The online version of this article includes the following figure supplement(s) for figure 1:

**Figure supplement 1.** AlphaFold-predicted[31,32] structure of human TRMT1.

in 2021 (*Suthar et al., 2022*; *Rahmani et al., 2022*; *Watson et al., 2022*; *Mohammed et al., 2022*), but a fundamental understanding of coronavirus host protein interactions and biology continues to be an important goal to inform ongoing and future therapeutic design. The SARS-CoV-2 main protease (M$^{pro}$ or Nsp5) is a well-studied antiviral drug target because its activity is essential for viral replication. More than 600 M$^{pro}$ crystal structures – the majority with bound small molecule inhibitors – are currently available in the protein databank (*Burley et al., 2021*). M$^{pro}$ is necessary for the proteolysis of 11 different cleavage sites in the two SARS-CoV-2 extended viral polypeptides, which results in the liberation of mature non-structural proteins that are essential for host infection and viral propagation (*Wilamowski et al., 2021*; *Yan et al., 2020*; *Shin et al., 2020*; *Zheng et al., 2021*; *Thoms et al., 2020*; *Yuan et al., 2020*). Approximately 20 structures of M$^{pro}$ in complex with different viral cleavage sequences have been determined (*MacDonald et al., 2021*; *Zhao et al., 2022*; *Shaqra et al., 2022*), in which the M$^{pro}$ homodimer has a peptide substrate bound to its active site (*Figure 1A*). However, a detailed understanding of SARS-CoV-2 M$^{pro}$ substrate selectivity and cleavage efficiency remains poorly defined, and the structural characterization of interactions between the viral protease and human host protein targets remains underexplored (*Hameedi et al., 2022*).

Early in the COVID-19 pandemic, Gordon et al. expressed tagged SARS-CoV-2 proteins in human embryonic kidney cells (HEK293T/17) and employed affinity purification mass spectrometry (AP-MS) to map proteome-wide virus-host protein interactions, in which a putative interaction between catalytically inactive SARS-CoV-2 M$^{pro}$ (Cys145Ala) and the human tRNA methyltransferase TRMT1 was identified (*Gordon et al., 2020b*). Analogous AP-MS experiments with wild-type (WT) M$^{pro}$ found no stable interaction with TRMT1, and subsequent mapping with SARS-CoV-1 proteins also found a TRMT1 interaction exclusively with Cys145Ala M$^{pro}$, suggesting that coronavirus M$^{pro}$ may recognize and actively cleave human TRMT1 in cells (*Gordon et al., 2020b*; *Gordon et al., 2020a*). TRMT1 is a tRNA-modifying enzyme responsible for installing $N^2,N^2$-dimethylguanosine (m2,2G), an abundant

tRNA modification found at the G26 position of human tRNAs (*Chou et al., 2017*). Human HEK293T cells lacking TRMT1, and therefore the m2,2G26 modification, have significantly decreased global protein synthesis and reduced proliferation (*Dewe et al., 2017*). Human neural stem cells with TRMT1 knockdowns were found to have hypersensitivity to redox stress, implicating TRMT1 and the m2,2G26 modification in the regulation of redox homeostasis (*Dewe et al., 2017*). TRMT1 contains a peptide sequence (527–534) consistent with the cleavage consensus for M$^{pro}$, located in an AlphaFold2-predicted (*Varadi et al., 2022*; *Jumper et al., 2021*) surface-exposed region (*Figure 1—figure supplement 1*) linking the TRMT1 N-terminal SAM-dependent methyltransferase domain and the C-terminal zinc finger domain (*Figure 1B*). SARS-CoV-2 M$^{pro}$-directed cleavage of TRMT1, and subsequent downregulation of tRNA m2,2G26 modification during infection, could therefore have direct impacts on both host and viral protein synthesis, as well as phenotypes linked to redox stress.

Here, we show that SARS-CoV-2 M$^{pro}$ proteolyzes endogenous human TRMT1 in human cell lysate and cleaves an amino acid sequence located between methyltransferase and zinc finger domains in TRMT1 with similar kinetic parameters to known viral polypeptide cleavage sites. M$^{pro}$-mediated cleavage of TRMT1 results in reduced tRNA binding affinity and the complete loss of its tRNA methyltransferase activity in vitro. We have determined the structure of M$^{pro}$ in complex with a TRMT1 peptide that shows how the viral protease recognizes and cleaves this human protein sequence and reveals a unique binding mode for the TRMT1 peptide in the M$^{pro}$ active site. Further kinetic analysis and molecular dynamics (MD) simulations show how the flexible M$^{pro}$ active site can accommodate diverse peptide geometries and suggest that kinetic discrimination between substrate sequences occurs during later steps in the M$^{pro}$-catalyzed cleavage reaction. Like many viruses, SARS-CoV-2 hijacks and disrupts diverse host biochemical pathways thereby promoting viral replication and avoiding host detection and immunological response. Recent studies have identified a large number of human proteins that may be targeted and cleaved by M$^{pro}$, many of which are involved in regulation of cellular proliferation, inflammatory response, transcription, translation, ubiquitination, apoptosis, and metabolism (*Miczi et al., 2020*; *Koudelka et al., 2021*; *Moustaqil et al., 2021*; *Yucel et al., 2022*; *Zhang et al., 2021*; *Pablos et al., 2021*; *Meyer et al., 2021*). Our biochemical and structural data show that TRMT1 can be targeted and cleaved by the SARS-CoV-2 main protease. SARS-CoVs employ multiple strategies to alter host translation (*Nakagawa et al., 2016*; *Gorkhali et al., 2021*), and cleavage of TRMT1 and corresponding loss of the tRNA m2,2G26 modification may be another mechanism by which the virus is able to disrupt or manipulate regulation of host protein synthesis. Furthermore, it is possible that proteolysis of TRMT1 during SARS-CoV-2 infection could contribute to some cellular pathogenesis observed with COVID-19.

## Results

### Full-length TRMT1 is cleaved by SARS-CoV-2 M$^{pro}$ in vitro

SARS-CoV-1 and SARS-CoV-2 proteome-wide virus-host protein interaction maps identified a putative interaction between catalytically inactive M$^{pro}$ (Cys145Ala) and human TRMT1, but no such stable interaction was found using WT protease, suggesting M$^{pro}$ may target and cleave TRMT1 in cells (*Gordon et al., 2020b*; *Gordon et al., 2020a*). A peptide sequence (527–534) located between the methyltransferase domain and zinc finger domain of human TRMT1 fits the cleavage consensus sequence of SARS-CoV-2 M$^{pro}$ (*Figure 1B*). To determine if full-length (FL) TRMT1 is susceptible to cleavage by M$^{pro}$, we first measured proteolysis of human TRMT1 expressed and purified from *E. coli* by recombinant purified SARS-CoV-2 M$^{pro}$ (*Figure 2A*). TRMT1 was incubated with either catalytically inactive M$^{pro}$ Cys145Ala or active WT M$^{pro}$, and TRMT1 cleavage was monitored by western blot using two TRMT1-specific polyclonal antibodies: a dual-domain recognizing antibody specific for portions of both the methyltransferase and zinc finger domain (anti-TRMT1 460–659), and a single-domain recognizing antibody specific for only the zinc finger domain (anti-TRMT1 609–659). As expected, FL TRMT1 (~75 kDa) was stable during a 2 hr incubation with inactive Cys145Ala M$^{pro}$. However, when incubated with WT M$^{pro}$, the intensity of the band corresponding to FL TRMT1, as measured by both TRMT1 antibodies, was reduced by >90% after 2 hr. In western blots with cleaved TRMT1, the dual domain-recognizing anti-TRMT1(460–659) antibody showed the appearance of two new lower molecular weight bands, corresponding to the anticipated molecular weights of the M$^{pro}$-mediated TRMT1 cleavage products (~61 kDa TRMT1 methyltransferase domain and ~14 kDa TRMT1 zinc finger

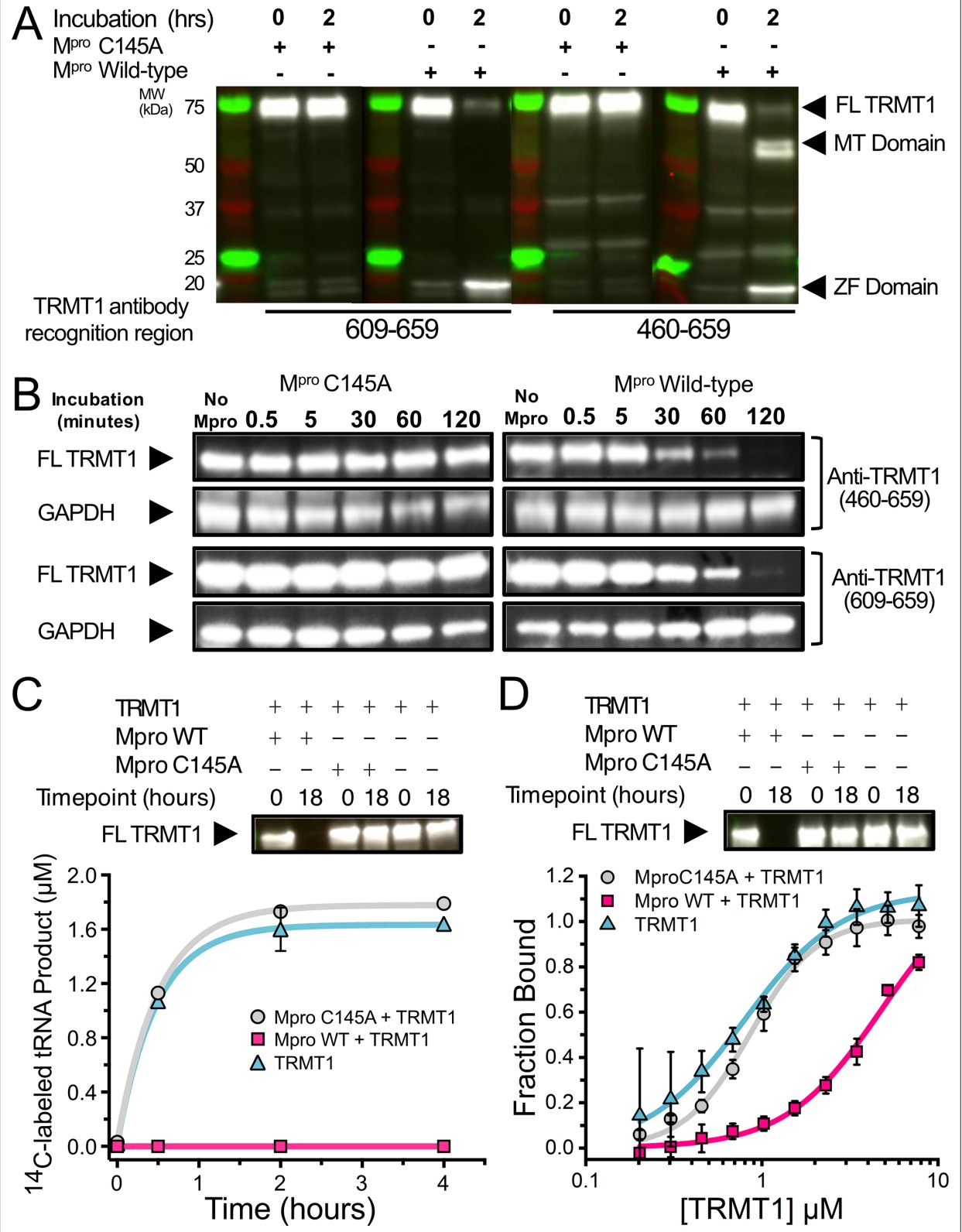

**Figure 2.** SARS-CoV-2 M^pro cleaves full-length (FL) human TRMT1 which impacts methyltransferase activity and tRNA binding. (**A**) Western blots of recombinantly purified FL TRMT1 incubated with 10 µM catalytically inactive (Cys145Ala) or active (wild-type [WT]) SARS-CoV-2 M^pro at 37°C. Incubation with WT M^pro results in proteolysis of FL TRMT1 and the appearance of cleavage products corresponding the zinc finger (ZF) domain (observed with both anti-TRMT1 (609–659) and anti-TRMT1 (460–659) antibodies) and the methyltransferase (MTase) domain (observed with only anti-TRMT1 (460–659)

*Figure 2 continued on next page*

*Figure 2 continued*

antibody). (**B**) Western blots of endogenous human TRMT1 in HEK293T cell lysate incubated with 10 µM of either catalytically inactive (Cys145Ala) or active (WT) M$^{pro}$ at 37°C. Endogenous FL TRMT1 is stable in human cell lysate over the course of a 2 hr incubation with C145A M$^{pro}$ (left) and is rapidly proteolyzed upon incubation with WT M$^{pro}$ (right). GAPDH was stained in conjunction with TRMT1 antibodies and used as a loading control. (**C**) Recombinant, FL TRMT1 cleaved for 18 hr with WT M$^{pro}$ (complete cleavage confirmed by western blot using anti-TRMT1 (460–659), top panel), has no observable methyltransferase activity on a human FL tRNA$^{phe}$ substrate (bottom panel, pink squares); in contrast, robust tRNA methylation activity is still seen with TRMT1 incubated for 18 hr with M$^{pro}$ Cys145Ala or no protease (bottom panel, gray circles and blue triangles, respectively). TRMT1 tRNA methyltransferase activity was measured by monitoring radiolabel incorporation into tRNA substrate in reactions with cofactor $^{14}$C-SAM. (**D**) Recombinant, FL TRMT1 cleaved for 18 hr with WT M$^{pro}$ (complete cleavage confirmed by western blot using anti-TRMT1 (460–659), top panel), has reduced binding affinity (~6-fold change) for human tRNA$^{Phe}$ (bottom panel, pink squares) compared to uncleaved TRMT1 that had been incubated for 18 hr with either M$^{pro}$ Cys145Ala or no protease (bottom panel, gray circles and blue triangles, respectively). TRMT1-tRNA binding was measured by electrophoretic mobility shift assay (EMSA), where bound and unbound tRNA species at different TRMT1 concentrations were separated, visualized by SYBR Gold staining, and quantified using ImageJ to obtain fraction bound values. Methyltransferase and binding assays in (**C, D**) were carried out in triplicate and errors are shown as SEM; fitted kinetic and binding parameters are shown in *Figure 2—source data 1*.

The online version of this article includes the following source data and figure supplement(s) for figure 2:

**Source data 1.** Table of methyltransferase activity ($k_{obs}$) and tRNA binding affinity ($K_D$) parameters for *Figure 2C and D*.

**Source data 2.** Raw, uncropped immunoblots for gel images in *Figure 2*.

**Source data 3.** Annotated, uncropped immunoblots for gel images in *Figure 2*.

**Source data 4.** Primary kinetic and binding data for *Figure 2C and D*.

**Figure supplement 1.** TRMT1 binding to tRNA$^{phe}$ substrate was measured using an electrophoretic mobility shift assay (EMSA).

**Figure supplement 1—source data 1.** Raw, uncropped EMSA gel images for *Figure 2—figure supplement 1*.

**Figure supplement 1—source data 2.** Annotated, uncropped EMSA gel images for *Figure 2—figure supplement 1*.

domain). After demonstrating that recombinant FL TRMT1 isolated from *E. coli* could be cleaved by SARS-CoV-2 M$^{pro}$, we next asked whether endogenous TRMT1 from human cells could be similarly cleaved by the viral protease. HEK293T cell lysate was incubated with either Cys145Ala or WT M$^{pro}$, and endogenous TRMT1 levels were monitored over time by western blot (*Figure 2B*). Incubation with Cys145Ala M$^{pro}$ resulted in no change to the FL TRMT1 band over the course of 2 hr, while incubation with WT M$^{pro}$ resulted in time-dependent proteolysis of FL TRMT1. Unlike in cleavage assays with recombinant TRMT1, no build-up of cleavage products was observed during M$^{pro}$-mediated proteolysis of endogenous TRMT1, suggesting instability and rapid degradation of cleaved TRMT1 fragments in human cell lysate. Together, these experiments demonstrate that both recombinant FL TRMT1 and endogenous FL TRMT1 in human cell lysate are viable substrates for cleavage by SARS-CoV-2 M$^{pro}$.

## Cleavage of TRMT1 results in complete loss of tRNA m2,2G modification activity and reduced tRNA binding in vitro

Next, we explored the consequences of M$^{pro}$-mediated cleavage of TRMT1 on both tRNA m2,2G modification and tRNA binding. TRMT1 was incubated overnight alone, with WT M$^{pro}$ (to generate a fully cleaved TRMT1 product), or with M$^{pro}$ Cys145Ala (no cleavage control). The M$^{pro}$ cleavage reaction was monitored by western blot using anti-TRMT1 (460–659) antibody to confirm the depletion of FL TRMT1 by M$^{pro}$ WT (*Figure 2C and D*, top). We then measured the impact of TRMT1 cleavage on its tRNA modifying activity by radiolabel-based methyltransferase assays with S-[methyl-$^{14}$C]-adenosyl methionine ($^{14}$C-SAM; *Figure 2C*, bottom) and its tRNA binding affinity by electrophoretic mobility shift assays (EMSAs) with FL human tRNA$^{phe}$ (*Figure 2D*, bottom). Cleaved TRMT1 generated by incubation with M$^{pro}$ WT exhibits a total loss of measurable tRNA methyltransferase activity, compared to TRMT1 incubated with M$^{pro}$ Cys145Ala, which retains full activity comparable to TRMT1 alone (*Figure 2C*, *Figure 2—source data 1*). Similarly, we found that M$^{pro}$-cleaved TRMT1 also had reduced affinity for tRNA, whereas TRMT1 incubated with M$^{pro}$ Cys145Ala had comparable tRNA binding affinity to TRMT1 alone (*Figure 2D*, *Figure 2—source data 1*). While cleaved TRMT1 still retained some affinity for tRNA, we observed in EMSA experiments that cleaved TRMT1-tRNA complexes migrate at significantly higher apparent molecular weights as compared to uncleaved TRMT1-tRNA complexes (*Figure 2—figure supplement 1*). This suggests that the TRMT1 cleavage products in vitro may form oligomers that have reduced affinity for tRNA, but based on the methyltransferase activity assays above, these oligomeric TRMT1-tRNA complexes cannot efficiently methylate bound tRNA.

These observations are consistent with other studies that have shown truncated TRMT1 constructs lacking the zinc finger domain have reduced tRNA modification activity in cells (*Dewe et al., 2017*) or in vitro (*Lu and Zhou, 2023*). Together, our in vitro tRNA binding and modification assays directly show that M^pro-mediated cleavage of TRMT1 reduces its substrate tRNA binding affinity and completely incapacitates its tRNA m2,2G modification catalysis.

## Structure of TRMT1 peptide bound to the M^pro active site

To visualize how TRMT1 is recognized and cleaved by the viral protease active site, we determined the co-crystal structure of catalytically inactive Cys145Ala mutant M^pro in complex with the TRMT1 peptide sequence corresponding to the expected cleavage site, human TRMT1 residues 526–536 (*Figure 3*, *Figure 3—source data 1*). The 1.9 Å resolution structure shows clear density for the TRMT1 peptide bound to the active site in one protomer of the M^pro dimer (*Figure 3A*, *Figure 3—figure supplement 1A*). As expected based on alignment with the M^pro consensus cleavage sequence, TRMT1 Gln530 occupies the critical S1 subsite pocket in the M^pro active site (*Figure 3A*), positioning the Gln530-Ala531 peptide bond directly adjacent to the His41/Cys145 catalytic dyad for proteolytic cleavage (*Figure 3B*). TRMT1 Leu529 corresponds to substrate position P2 and occupies the hydrophobic M^pro S2 pocket (*Figure 3A*), packing against M^pro residues His41, Met49, Met165, and Gln189 (*Figure 3—figure supplement 1B*). P3′ substrate residue TRMT1 Phe533 occupies the M^pro S3′ pocket (*Figure 3A*), which is composed by M^pro residues His41, Cys44, Met49, and the backbone of Thr45 (*Figure 3—figure supplement 1C*). TRMT1 peptide binding to M^pro is also mediated by numerous key sidechain and mainchain hydrogen bonding contacts (*Figure 3C and D*). The TRMT1 P1 Gln530 sidechain is specifically recognized by hydrogen bonding interactions with M^pro residues Phe140 (*Figure 3C*) and His163 (*Figure 3D*), consistent with previous structures of M^pro bound to viral peptide substrates (*Shaqra et al., 2022*). Additionally, the TRMT1 P2′ Asn532 sidechain interacts with M^pro Asn142 (*Figure 3C*). A number of TRMT1 peptide backbone atoms (from TRMT1 residues Arg528, Leu529, Gln530, Asn532, and Thr534) make hydrogen bond contacts with M^pro surface residues (M^pro Gly143, Glu166, and Gln189 contacts shown in *Figure 3C*; M^pro Thr24, Thr26, Cys145Ala, and His164 contacts shown in *Figure 3D*). The identified M^pro-targeted residues in human TRMT1 are conserved in the human population (i.e. no missense polymorphisms), showing that human TRMT1 can be recognized and cleaved by SARS-CoV-2 M^pro.

## TRMT1 engages the M^pro active site in a distinct binding conformation

M^pro has more than 10 native viral polypeptide substrates with relatively high sequence variability outside the conserved P1 Gln and the adjacent P2 and P1′ residues. We compared our human TRMT1-bound M^pro structure with available SARS-CoV-2 M^pro structures bound to known viral polypeptide cleavage sequences (*Figure 4A*; nsp4/5, nsp5/6, nsp6/7, nsp8/9, nsp9/10, nsp10/11, and nsp15/16). The overall structures of M^pro are very similar for all peptide-bound structures, with M^pro backbone RMSD values all below 1.6 (*Figure 4—figure supplement 1*). Analysis of aligned peptide substrates in the M^pro active site shows that both TRMT1 and viral peptide backbones have nearly identical conformations for the N-terminal P4 → P1′ residues. In contrast, backbone geometries of C-terminal peptide residues (P2′ → P4′) diverge more significantly and fall into two distinct binding modes distinguished primarily by the P2′ Ψ dihedral angle (*Figure 4B and C*). The majority of viral peptide substrates adopt a binding conformation with P2′ Ψ ≥157 ° in which the P3′ sidechain is positioned away from the M^pro surface (*Figure 4D*), which we have designated as the 'P3′-out' conformation. In contrast, TRMT1 and a single viral peptide substrate, nsp6/7, bind in a distinct conformation with P2′ Ψ ≤116 ° (*Figure 4C*) in which the P3′ sidechain (Phe in TRMT1, Met in nsp6/7) is positioned toward the M^pro surface where it displaces M^pro Met49 to open and occupy pocket S3′ (*Figure 4D*). We have designated this uncommon binding mode as the 'P3′-in' conformation.

## M^pro cleaves TRMT1 peptide with comparable kinetics to known viral polypeptide cleavage sites

To further examine whether TRMT1 is a viable substrate for SARS-CoV-2 M^pro, we compared kinetic parameters for proteolysis of the TRMT1 (526–536) peptide with the nsp4/5 and nsp8/9 viral polypeptide cleavage sequences, using fluorogenic peptide cleavage assays. The TRMT1 peptide is cleaved noticeably slower than the nsp4/5 N-terminal auto-processing sequence (*Figure 5A*, *Figure*

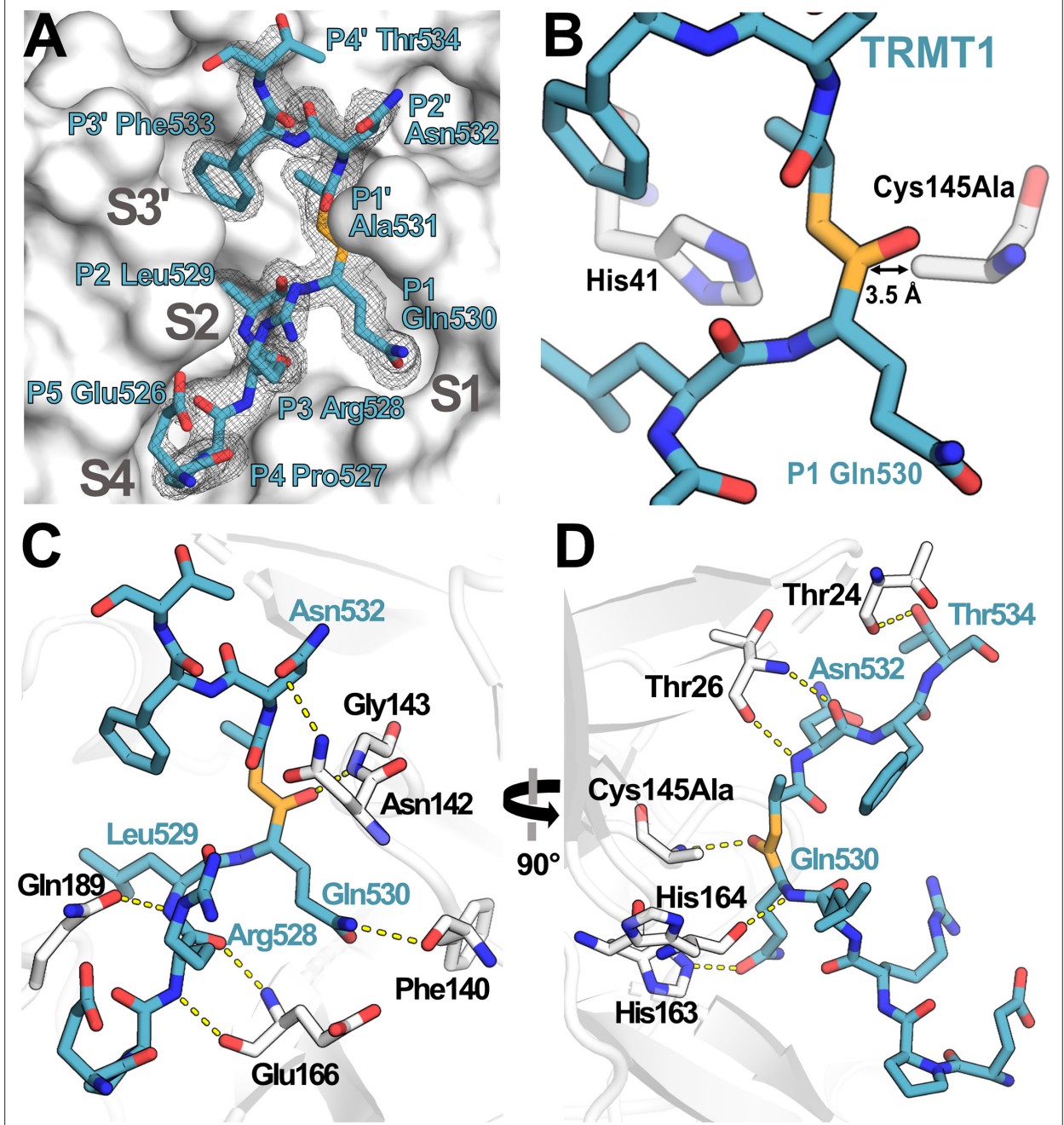

**Figure 3.** Structure of human TRMT1 (526–536) peptide bound to SARS-CoV-2 M^pro. (**A**) TRMT1 peptide (blue) bound in M^pro active site (gray) showing substrate binding pockets S1, S2, S4, and S3′. $F_o$-$F_c$ omit electron density map of TRMT1 peptide bound to M^pro is shown contoured at 1σ. TRMT1 Gln P1, an ultra-conserved residue in the M^pro cleavage consensus which is critical for M^pro-mediated proteolysis, is nestled in the S1 pocket of the M^pro active site. The scissile P1 – P1′ amide linkage of TRMT1 is colored orange. (**B**) The TRMT1 P1 Gln amide is positioned for cleavage near M^pro catalytic dyad residues His41 and Cys145Ala in the protease active site. (**C, D**) Direct hydrogen bond contacts formed between M^pro residues (white) and the bound TRMT1 peptide (light blue) are illustrated as yellow dashed lines; (C and D) are views rotated by 90°, highlighting different TRMT1-M^pro hydrogen bonding interactions. M^pro Phe140 and His163 recognize the TRMT1 P1 Gln530 sidechain; TRMT1 Asn532 and M^pro Asn142 engage in sidechain-sidechain interaction; additional backbone hydrogen bond contacts include M^pro Thr24-TRMT1 Thr534, M^pro Thr26-TRMT1 Asn532, M^pro Asn142-TRMT1 Asn532, M^pro Glu166-TRMT1 Arg528, and M^pro Gln189-TRMT1 Leu529; many of these interactions are consistent with canonical M^pro-peptide substrate contacts in the active site.

The online version of this article includes the following source data and figure supplement(s) for figure 3:

**Source data 1.** Table of data and refinement statistics for M^pro-TRMT1 crystal structure.

*Figure 3 continued on next page*

*Figure 3 continued*

**Figure supplement 1.** The M^pro-TRMT1 (526–536) co-crystal structure uncovers how TRMT1 peptide residues occupy the M^pro active site substrate binding pockets.

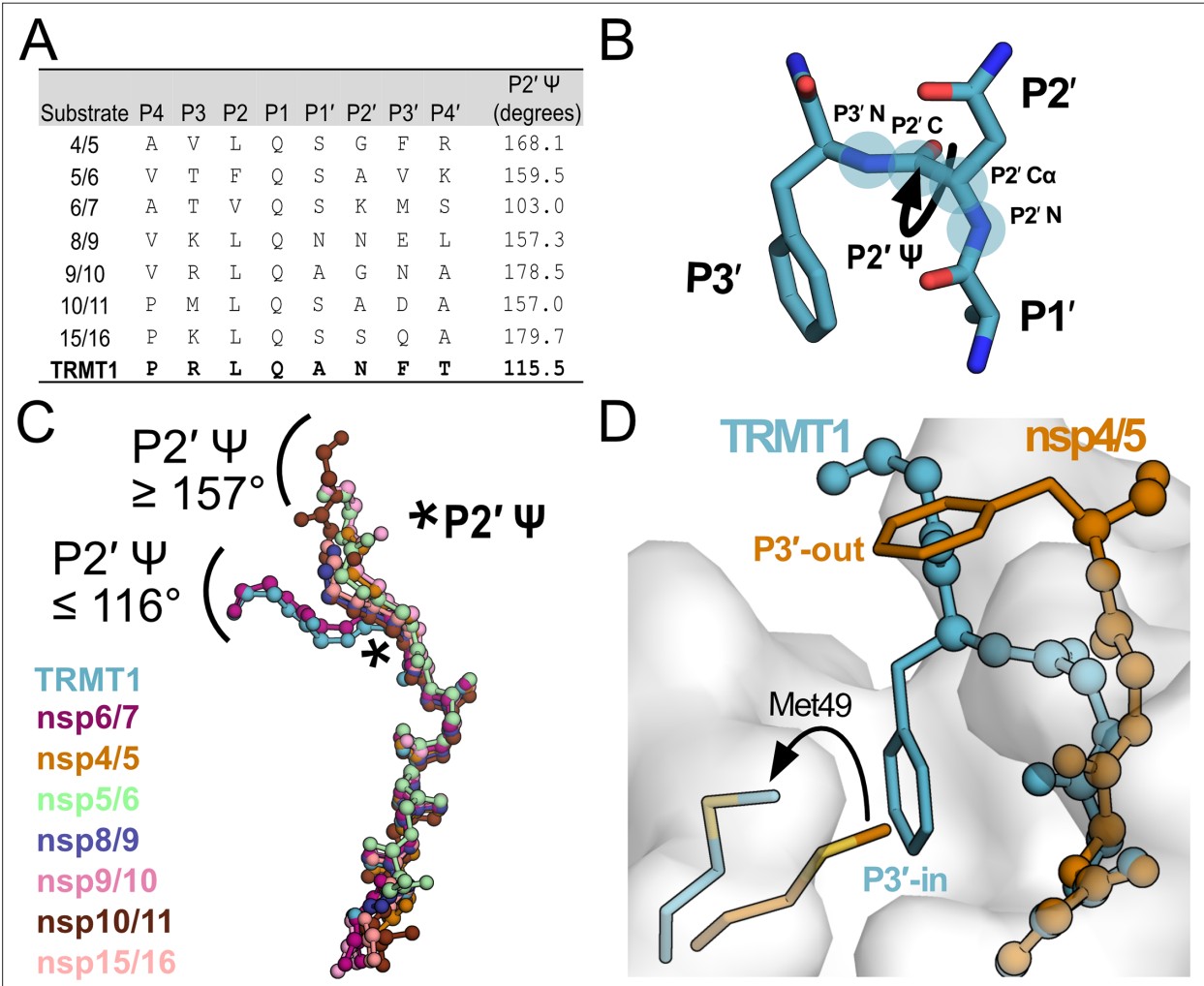

**Figure 4.** M^pro-peptide structures illustrate two distinct substrate binding modes, P3'-in and P3'-out. (**A**) Comparison of known M^pro substrate cleavage sequences and the P2' Ψ backbone dihedral angles measured in the corresponding C145A M^pro-peptide structures for each substrate. We included all known C145A M^pro-viral peptide structures in this analysis, except those that were missing the P3' residue or had poorly defined electron density for the C-terminal portion of the peptide; structures used in this analysis are PDB IDs: 7MGS, 7T8M, 7DVW, 7T9Y, 7TA4, 7TA7, 7TC4, and 9DW6. Additionally, since a C145A M^pro-nsp6/7 structure was not available, we used an H41A M^pro-nsp6/7 structure (PDB 7VDX) for this analysis. (**B**) Section of an M^pro-bound peptide substrate showing residues P1', P2', and P3', with the key P2' Ψ dihedral angle illustrated with a curved arrow; the four backbone atoms that define the P2' Ψ dihedral angle are labeled and highlighted with blue circles (P2'N–P2'Cα–P2'C–P3'N). (**C**) Alignment of peptide substrate backbones in the M^pro active site reveals two distinct binding modes at the C-terminal end of the bound peptides characterized by P2' Ψ dihedral angles ≥157° (nsp4/5, nsp5/6, nsp8/9, nsp9/10, nsp10/11, nsp15/16) or ≤116° (TRMT1, nsp6/7). Peptide overlays were generated by aligning SARS-CoV-2 M^pro-peptide substrate structures in PyMOL. The location of the P2' Ψ dihedral angle in the substrate peptide backbone is denoted with a star. (**D**) Alignment of nsp4/5- and TRMT1-bound M^pro structures showing divergent C-terminal peptide substrate binding modes in the M^pro active site. The backbone geometry of nsp4/5 (P2' Ψ =168°) positions the P3' Phe sidechain away from the M^pro surface ('P3'-out' conformation), while the TRMT1 backbone geometry (P2' Ψ =115°) positions the P3' Phe sidechain toward the M^pro active ('P3'-in' conformation) site where it displaces M^pro Met49 to open and occupy the S3' pocket.

The online version of this article includes the following figure supplement(s) for figure 4:

**Figure supplement 1.** An alignment of M^pro structures for each M^pro-peptide complex used in the analysis shown in *Figure 4A and C* (top).

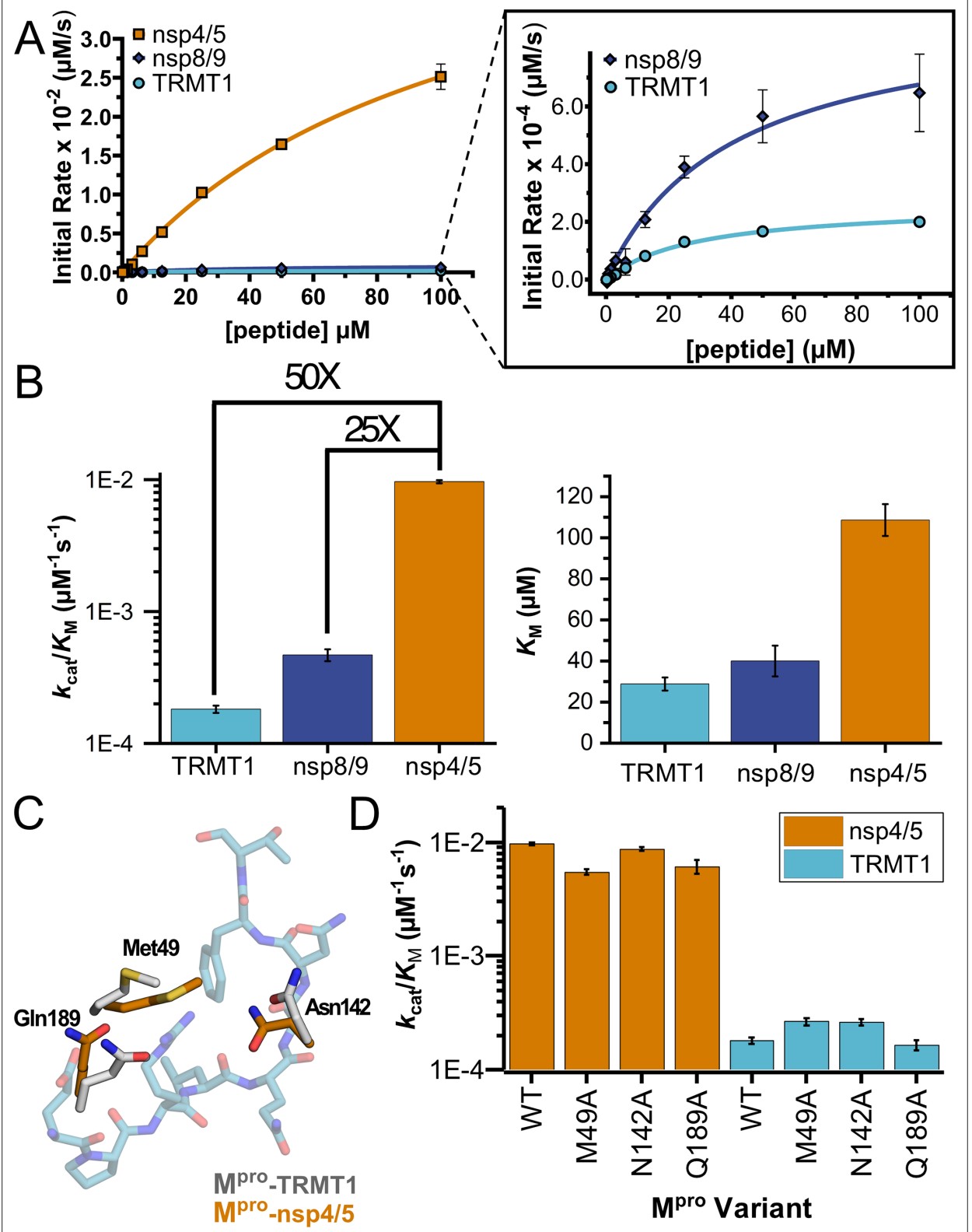

**Figure 5.** Human TRMT1 peptides are cleaved with similar catalytic efficiencies to known M^pro substrates. (**A**) Kinetics of nsp4/5, nsp8/9, and TRMT1 peptide cleavage by M^pro. To initiate the reaction, 50 nM enzyme was added to 100–0.097 μM peptide. Each fluorogenic peptide was conjugated with a quenching moiety, and upon peptide cleavage, the fluorescence of the cleavage product was measured to determine initial rates of the reaction. Nsp4/5 cleavage rates were faster than those observed for the nsp8/9 or TRMT1 peptides, but nsp8/9 and TRMT1 sequences exhibit similar M^pro-

*Figure 5 continued on next page*

*Figure 5 continued*

mediated cleavage rates. (**B**) The catalytic efficiency ($k_{cat}/K_M$) of TRMT1 peptide cleavage by M^pro is similar to that for nsp8/9 peptide cleavage; both of these substrates are cleaved significantly slower than the nsp4/5 sequence. This suggests that TRMT1 is a feasible substrate for M^pro. (**C**) Illustration of changes in M^pro Met49, Asn142, and Gln189 residue positioning in TRMT1-bound (white) vs nsp4/5-bound (orange) structures. The TRMT1 peptide is shown in blue; nsp4/5 peptide is not shown. (**D**) No major changes in catalytic efficiency are observed for nsp4/5 and TRMT1 peptide cleavage upon mutagenesis of key M^pro residues involved in TRMT1 binding and recognition. All kinetic assays were carried out in triplicate and errors are shown as SEM.

The online version of this article includes the following source data and figure supplement(s) for figure 5:

**Source data 1.** Table of kinetic parameters associated with *Figure 5*.

**Source data 2.** Primary fluorescence data and calibration and correction calculations used to determine kinetic parameters in *Figure 5*.

**Figure supplement 1.** Analysis of a point mutation to P1′ Ala residue of TRMT1 (526–536).

*5—source data 1*), with an approximately 200-fold decrease in $k_{cat}$, 4-fold decrease in $K_M$, and 50-fold decrease in catalytic efficiency ($k_{cat}/K_M$) compared to M^pro-mediated cleavage of the nsp4/5 peptide (*Figure 5B*, *Figure 5—source data 1*). However, TRMT1 is cleaved with very comparable kinetics to the viral nsp8/9 cleavage sequence, with $k_{cat}$, $K_M$, and $k_{cat}/K_M$ values all within ~3-fold for TRMT1 vs nsp8/9 peptide cleavage. These kinetic experiments show that, while the TRMT1 M^pro target sequence is cleaved less efficiently than the canonical nsp4/5 auto-processing site, the TRMT1 sequence is cleaved with kinetics very similar to the known nsp8/9 M^pro cleavage site in the viral polypeptide. This suggests that TRMT1 (526–536) is a viable substrate for M^pro-targeted proteolysis.

## Residues involved in M^pro-TRMT1 recognition have only small effects on cleavage efficiency

We next compared the M^pro-peptide interactions observed in TRMT1-, nsp4/5-, and nsp8/9-bound structures and identified several M^pro residues involved in direct substrate contacts that differ between the TRMT1- and nsp4/5- or nsp8/9-bound structures (*Figure 5C*). In the TRMT1-bound structure, M^pro Met49 is shifted to accommodate TRMT1 P3′ substrate residue Phe533 binding in the S3′ pocket, and the M^pro Asn142 and Gln189 sidechains are repositioned to form hydrogen bonds with the TRMT1 peptide backbone. To test how these M^pro active site residues affect TRMT1 recognition and catalysis, we generated single alanine point mutations at each of these three sites and measured kinetic parameters for M^pro-mediated proteolysis of nsp4/5 and TRMT1 peptides. Additionally, we mutated the P1′ TRMT1 Ala531 to Ser, the P1′ residue found in nsp4/5 and nearly all of the other M^pro-targeted viral cleavage sequences, to test the effects of mutating a key position in the substrate peptide. Surprisingly, we saw no substantial changes in proteolysis kinetics for cleavage of either nsp4/5 or TRMT1 substrates with any of the tested M^pro mutants (Met49Ala, Asn142Ala, Gln189Ala) or the TRMT1 Ala531Ser mutant peptide (*Figure 5D*, *Figure 5—source data 1*, *Figure 5—figure supplement 1*), suggesting none of these residues alone play a significant role in determining differential cleavage rates for TRMT1 vs nsp4/5.

## Evolutionary insights into TRMT1's conserved M^pro cleavage site reveals M^pro proteolytic resistance in rodents

To determine whether the TRMT1 sequence at the M^pro cleavage site is unique to human TRMT1 and whether TRMT1 bears signatures of molecular arms-races with pathogens during mammalian evolution, we performed phylogenetic and positive selection analyses (*Tenthorey et al., 2022*; *Sironi et al., 2015*). Through an evolutionary screen, *Cariou et al., 2022*, previously identified some signatures of rapid evolution in primate TRMT1, potentially driven by adaptation to ancient viral pathogens, including coronaviruses. Through comprehensive analyses, we identified rapidly evolving sites at the N- and C-termini of primate TRMT1, but we did not find any evidence of sites under positive selection at the M^pro-mediated TRMT1 (526–536) cleavage sequence (*Figure 6—figure supplement 1A*, *Figure 6—source data 1*). In fact, this sequence is highly conserved in primates (*Figure 6A*). A TRMT1 sequence analysis at the mammalian level further showed that the M^pro cleavage site of TRMT1 (residues 526–536) is highly conserved in most mammals, including in bats that are the reservoir of SARS-CoVs (*Figure 6B*). However, one exception is in rodents, where there has been a Q to K substitution fixed in all Muroidea (mouse, rat, hamster, etc.) (*Figure 6C*, *Figure 6—figure supplement 1B*). To test

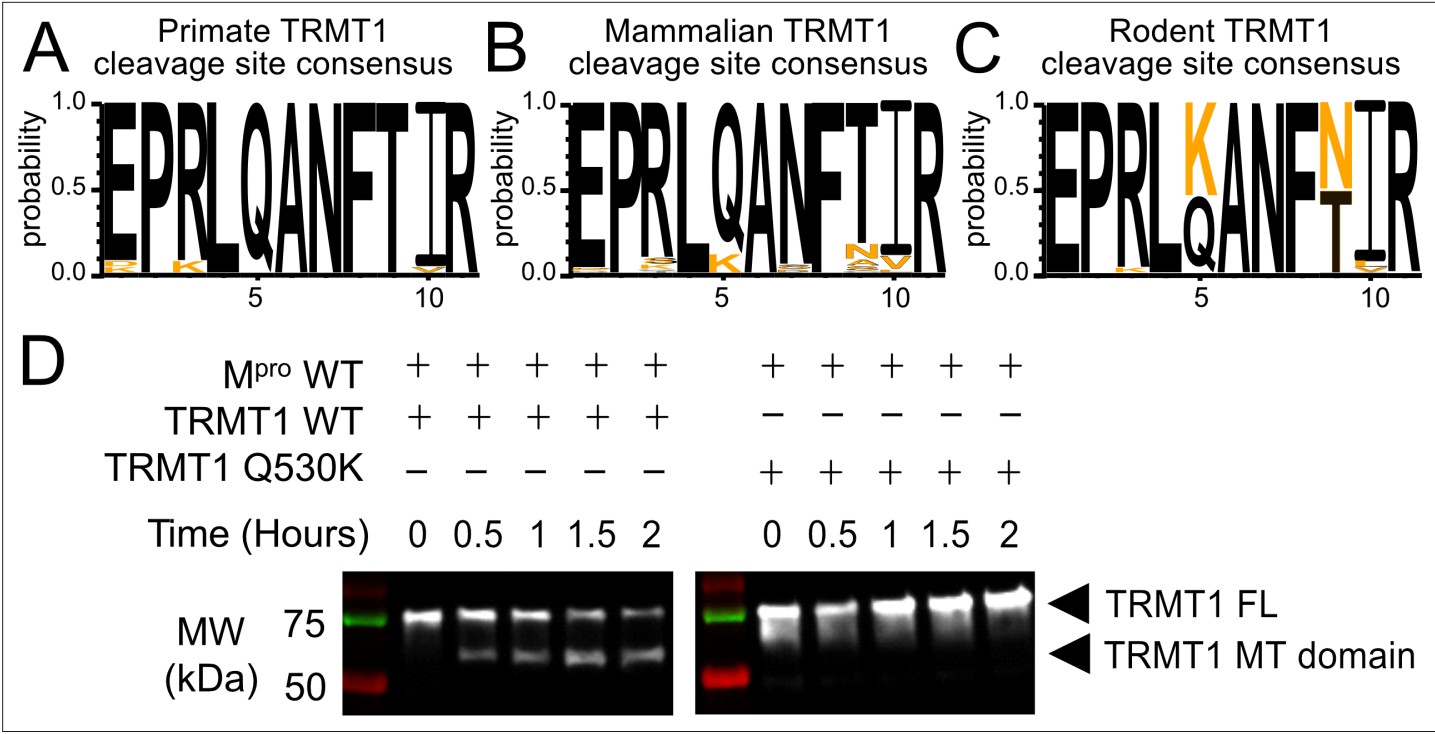

**Figure 6.** The M^pro^-targeted cleavage sequence is conserved in most mammalian TRMT1 proteins, except rodents where TRMT1 is resistant to cleavage. (**A–C**) The M^pro^-targeted TRMT1 cleavage site sequence (human TRMT1 residues 526–536) is highly conserved in primates (**A**) and most mammals (**B**), with the notable exception of rodents (**C**), where the glutamine Q530 residue most critical for M^pro^-directed cleavage is substituted to a lysine in Muroidea. Sequence logo plots of the cleavage site in TRMT1 (526–536) were produced with WebLogo3. The human reference sequence is in black and orange residues show the differences. (**D**) Wild-type (WT) human TRMT1 is cleaved over the course of a 2 hr incubation with M^pro^ (left western blot panel), whereas human TRMT1(Q530K), which has the Q to K mutation found in Muroidea, is entirely resistant to cleavage during a 2 hr incubation with M^pro^ (right western blot panel).

The online version of this article includes the following source data and figure supplement(s) for figure 6:

**Source data 1.** Data used for sequence analyses in *Figure 6A–C*.

**Source data 2.** Raw, uncropped immunoblots for *Figure 6D*.

**Source data 3.** Annotated, uncropped immunoblots for *Figure 6D*.

**Figure supplement 1.** Evolution of mammalian TRMT1.

the impact of this substitution on M^pro^-mediated cleavage, we generated a TRMT1 Gln530Lys mutant and incubated it with M^pro^ WT for a period of 2 hr (*Figure 6D*). We found that TRMT1 Gln530Lys was resistant to M^pro^ cleavage and did not observe the formation of cleavage products (e.g. TRMT1 MT domain) compared to cleavage of TRMT1 WT under the same proteolytic conditions. This is consistent with a more recent report showing that mouse or hamster TRMT1 is not cleaved by M^pro^ when these enzymes are co-expressed in cells (*Lu and Zhou, 2023*). Our results confirm that the TRMT1 Gln530Lys substitution at the invariant P1 glutamine residue is sufficient to completely prevent TRMT1 proteolysis in vitro, strongly suggesting this mutation would prevent SARS-CoV M^pro^-directed TRMT1 cleavage during Muroidea infection.

## MD simulations suggest kinetic discrimination happens during later steps of M^pro^-catalyzed substrate cleavage

To further support the structural and biochemical data described above and attempt to understand how peptide substrate binding geometry may or may not be linked to M^pro^-catalyzed cleavage efficiency, we carried out MD simulations using nsp4/5-, nsp8/9-, and TRMT1-bound M^pro^ complexes. MD simulations show TRMT1 is stably bound to the M^pro^ active site and primarily adopts the P3'-in conformation observed in the crystal structure, where TRMT1 Phe533 occupies the S3' pocket with a probability of ~65% over the course of the 100 ns simulation (*Figure 7A*). In contrast, simulations of

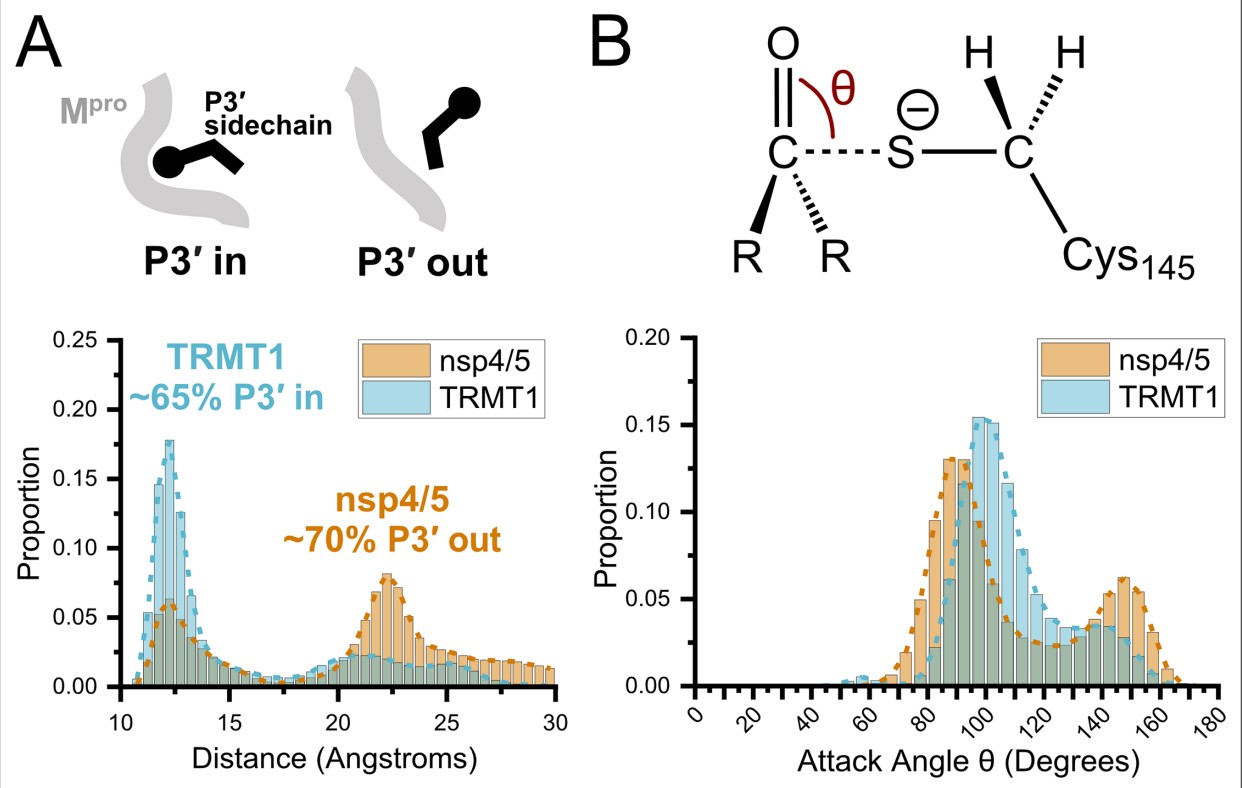

**Figure 7.** Molecular dynamics (MD) simulations confirm dominant peptide binding conformations and suggest discrimination in cleavage kinetics result from catalytic steps that follow initial binding and nucleophilic attack. (**A**) Distribution of the sum of the minimum distance for P3' Phe residue in nsp4/5 or TRMT1 from three residues (Thr25, Met49, Cys44) which form the S3' subsite; P3'-in and P3'-out conformations are illustrated above the distribution plot. The much larger proportion of TRMT1 at smaller distances reflects the peptide's preference for binding in the P3'-in conformation where TRMT1 P3' Phe occupies the S3' pocket during the majority of the MD simulation. (**B**) Distribution of the attack angle of the nucleophilic M$^{pro}$ Cys145 sulfur atom and the substrate carbonyl carbon atom in the to-be-cleaved amide bond (S–C=O angle $\theta$ , top illustration) during the course of the MD simulation. Although nsp4/5 has a higher proportion of attack angles observed closer to the optimal 90° compared to TRMT1, consistent with faster nsp4/5 cleavage kinetics, this small preference is insufficient to explain the 200-fold faster cleavage kinetics of nsp4/5 observed in experimental proteolysis assays.

The online version of this article includes the following source data and figure supplement(s) for figure 7:

**Source data 1.** Data from molecular dynamics (MD) simulations used to construct distribution plots in *Figure 7*.

**Figure supplement 1.** Substrate positioning at the M$^{pro}$ catalytic site does not readily explain observed differences in cleavage kinetics.

the nsp4/5-M$^{pro}$ complex show the nsp4/5 peptide primarily adopts the P3'-out conformation, where nsp4/5 Phe531 is oriented away from the M$^{pro}$ surface with ~70% probability, consistent with the observed nsp4/5 binding mode in the crystal structure. This computational analysis shows that while P3'-in and -out conformations can interconvert, the dominant binding modes for TRMT1 and nsp4/5 peptide substrates (P3'-in vs -out, respectively) are consistent across the crystal structures and MD simulations. Furthermore, because these conformations differ primarily at the C-terminal end of the peptide but have very similar geometries at the scissile amide bond, we find that the computationally and experimentally determined substrate binding poses fail to explain the large differences in cleavage kinetics observed for TRMT1 vs nsp4/5.

Previous structural and geometric analysis of diverse serine protease substrate-inhibitor complexes has shown that the consensus nucleophilic attack angle of the catalytic residue in the reactive Michaelis complex is approximately 90° (*Radisky, 2002*). We next asked whether differences in nucleophilic attack angle for the catalytic M$^{pro}$ Cys145 measured over the course of the MD simulation might be able to explain changes in M$^{pro}$-mediated cleavage kinetics. The distribution of the S–C=O nucleophilic attack angle shows that the M$^{pro}$-nsp4/5 complex has only ~9% increased probability to fall within 89±7°, as compared with the M$^{pro}$-TRMT1 complex over the course of the simulation (*Figure 7B*). Thus, while the nsp4/5 peptide shows slightly more favorable positioning of Cys145 for nucleophilic

attack in the Michaelis complex during the first step of cleavage, this is not nearly sufficient to explain the large difference (~200-fold in $k_{cat}$) in experimentally observed cleavage kinetics between nsp4/5 and TRMT1. Taken together with our TRMT1-M$^{pro}$ structure and kinetic analysis of M$^{pro}$ and TRMT1 mutants, these results strongly suggest that kinetic discrimination between peptide cleavage rates is likely to occur during a later step in the M$^{pro}$-catalyzed cleavage reaction that follows substrate binding, Michaelis complex formation, and initial nucleophilic attack, similar to some serine proteases (*Radisky, 2002*).

## Discussion

Recent studies mapping host-coronavirus protein interaction networks predicted high confidence interactions between the SARS-CoV-1 or SARS-CoV-2 main protease (M$^{pro}$) and the human tRNA-modifying enzyme TRMT1 (*Gordon et al., 2020b*; *Meyer et al., 2021*). In this work, we show that SARS-CoV-2 M$^{pro}$ can recognize and cleave the human TRMT1 526–536 sequence located between the TRMT1 methyltransferase and zinc finger domains and that cleavage of TRMT1 impairs tRNA substrate binding and abolishes tRNA m2,2G modification activity. M$^{pro}$ is able to efficiently cleave the TRMT1 526–536 sequence in recombinant FL TRMT1, as well as in endogenous TRMT1 from human cell lysate. The kinetic parameters determined for M$^{pro}$-mediated TRMT1 (526–536) peptide proteolysis are very similar to those measured for the known nsp8/9 viral polypeptide cleavage site. We also determined the structure of human TRMT1 (526–536) in complex with SARS-CoV-2 M$^{pro}$ C145A, which reveals a distinct binding mode for the TRMT1 substrate peptide that engages the M$^{pro}$ S3' pocket. Finally, though TRMT1 526–536 is highly conserved in primates and mammals, we found that Muroidea contains a critical substitution which confers M$^{pro}$ cleavage resistance. Our results show that the human tRNA-modifying enzyme TRMT1 is a viable substrate for the SARS-CoV-2 main protease and provide the structural basis for understanding TRMT1 recognition and proteolysis. Importantly, and concurrent with our own work, *Zhang et al., 2024*, report that human TRMT1 is cleaved during SARS-CoV-2 infection and that this leads to loss of m2,2G modification in virally infected cells. Together, our studies corroborate and characterize cleavage of human TRMT1 by SARS-CoV-2 M$^{pro}$ from the cellular to the atomic level and raise important new questions about the roles of tRNA methylation during viral infection.

Viruses have evolved diverse strategies that subvert host protein synthesis in order to downregulate host translation and optimize the translation of viral proteins (*de Breyne et al., 2020*; *Jan et al., 2016*; *Sharma et al., 2012*; *Walsh and Mohr, 2011*; *Roberts et al., 2009*). SARS-CoV-2 employs multiple mechanisms to disrupt host protein synthesis (*Thoms et al., 2020*; *Finkel et al., 2021*; *Hsu et al., 2021*; *Puray-Chavez et al., 2022*; *Eriani and Martin, 2022*; *Zhang et al., 2022*; *Banerjee et al., 2020*), and M$^{pro}$-mediated cleavage of human TRMT1 could contribute to the modulation of cellular translation during viral infection. TRMT1 is a tRNA methyltransferase whose m2,2G26 modification activity directly impacts global translational efficiency. Knockouts of TRMT1, or truncations of TRMT1 that remove its zinc finger domain, result in loss of the m2,2G26 tRNA modification and a significant decrease in translation levels in HEK293T cells (*Dewe et al., 2017*). Our results show that M$^{pro}$-mediated cleavage of TRMT1 results in the removal of the zinc finger domain, which weakens TRMT1-tRNA binding affinity and completely abolishes enzyme activity (*Figure 2C and D*). Additionally, TRMT1-deficient human cells, or cells with inactivating TRMT1 truncations, exhibit increased sensitivity to oxidative stress and links to neurological dysfunction (*Blaesius et al., 2018*; *Davarniya et al., 2015*; *Najmabadi et al., 2011*; *Zhang et al., 2020a*). Taken together, these observations suggest that M$^{pro}$-mediated cleavage of TRMT1 during SARS-CoV-2 infection may contribute to the downregulation of host translation and oxidative stress-related pathogenesis and phenotypes (*Cecchini and Cecchini, 2020*). Future experiments in different virally infected cell types will be required to definitively establish how M$^{pro}$-directed TRMT1 cleavage impacts cellular and viral translation, viral infectivity and propagation, and oxidative stress phenotypes during infection.

Our evolutionary analyses (*Figure 6*, *Figure 6—figure supplement 1*) raise several possibilities regarding the relationship of SARS-CoV and TRMT1. First, it may be that TRMT1 (526–536) is targeted for cleavage by M$^{pro}$ during coronavirus infection, but that this TRMT1 motif cannot easily mutate during evolution, due to structural constraints or essential function (*Tenthorey et al., 2022*; *Murrell et al., 2016*; *Abdul et al., 2018*; *Enard et al., 2016*), and therefore cannot evade viral proteolysis. Indeed, although the TRMT1 (526–536) cleavage sequence is found in a linker region between

structured domains, an AlphaFold2-predicted (*Varadi et al., 2022*; *Jumper et al., 2021*) TRMT1 structure suggests that many of these residues make contacts with the surface of the TRMT1 methyltransferase domain (*Figure 1—figure supplement 1*), which may be important for enzyme stability, domain orientation, or function. Second, it may be that TRMT1 cleavage by coronavirus M$^{pro}$ exerts insufficient adaptive pressure on the host. In this case, TRMT1 rapid evolution may be driven by other selective pressure or may be due to intrinsically disordered regions. Third, rapidly evolving sites in primate TRMT1 outside of the cleavage site may reflect evasion of SARS-CoV by providing structural escape through within-protein epistasis (*Starr and Thornton, 2016*). As coronaviruses infect multiple mammalian species, it would also be informative to our understanding of SARS-CoV evolution and function to further decipher Muroidea TRMT1 resistance to M$^{pro}$-mediated cleavage, asking if this resistance confers host advantages or disadvantages during coronavirus infection, and whether this change results from adaptation to ancient coronavirus infection.

Independent of the roles for human TRMT1 and m2,2G RNA modification during SARS-CoV-2 infection, our structural and kinetic results combined with computational analysis highlight distinct features of substrate-protease binding, recognition, and cleavage for SARS-CoV-2 M$^{pro}$. First, our TRMT1-M$^{pro}$ structure shows an uncommon peptide binding mode in which the P3′ TRMT1 Phe residue is buried in the S3′ pocket on the M$^{pro}$ surface ('P3′-in' conformation; *Figure 4D*). Of the currently known M$^{pro}$-peptide substrate structures, this unique M$^{pro}$ P3′-in binding mode is only seen for TRMT1 and nsp6/7, though notably the P3′-in conformation has also been observed in a crystal structure of SARS-CoV-1 M$^{pro}$ in complex with its C-terminal auto-processing site (VTFQGKFK), which like TRMT1, contains a phenylalanine at the P3′ site (*Muramatsu et al., 2016*). This unique binding mode demonstrates the flexible accommodation of diverse substrate binding poses in the M$^{pro}$ active site and highlights the availability of the S3′ pocket for inhibitor binding and therapeutic design. Second, consistent with previous mutagenesis studies of M$^{pro}$ (*Flynn et al., 2022*), we find that mutations to residues on the surface of M$^{pro}$ involved in direct binding and recognition of substrate, including Met49, Asn142, and Gln189, have surprisingly little impact on cleavage kinetics (*Figure 5D*). Similarly, mutating TRMT1 P1′ Ala531 to Ser (the P1′ residue in nsp4/5 and most other substrate sequences) had no effect on cleavage kinetics and MD simulations still predict this peptide will favor the P3′-in binding conformation (*Figure 5—figure supplement 1*). Third, comparing nsp4/5-, nsp8/9-, and TRMT1-bound M$^{pro}$ crystal structures, the scissile amide bond that links substrate residues P1 and P1′ are positioned almost identically in the active site and located at similar distances from the Cys145Ala residue (*Figure 7—figure supplement 1A*). Likewise, deviations away from 180° in the dihedral angle of the scissile amide bond in these M$^{pro}$-peptide structures, which could indicate ground state destabilization and help explain accelerated proteolysis for nsp4/5 (*Zhou et al., 2016*), are also inconsistent with the observed trends in peptide cleavage rates (*Figure 7—figure supplement 1B*). MD simulations of nsp4/5 and TRMT1 peptide substrates bound to M$^{pro}$ support the observation that the unique P3′-in conformation is favored for TRMT1 binding, but that only subtle differences in geometry are present at the nucleophilic Cys145 residue and scissile peptide bond in the M$^{pro}$-substrate complexes (*Figure 7*), which cannot account for the order of magnitude differences in observed cleavage kinetics between ns4/5 and TRMT1 substrates. Therefore, the large differences in catalytic efficiency observed for M$^{pro}$-mediated cleavage of nsp4/5 vs nsp8/9 and TRMT1 are not easily explained by structural analysis of the substrate-bound M$^{pro}$ structures or the M$^{pro}$-peptide geometries and conformations observed in the MD simulations.

Why do a small number of peptide substrate sequences favor the P3′-in conformation? How does substrate sequence or conformation impact cleavage efficiency? Taken together, our structural, kinetic, and computational data described above suggest that: (a) the conformational preferences of bound substrates are not clearly linked to simple sequence features, (b) mutations to individual residues on M$^{pro}$ or its peptide substrate that appear important for M$^{pro}$-substrate interactions or conformation are often insufficient to change binding conformation or cleavage kinetics, and therefore, (c) kinetic discrimination of substrate cleavage likely happens during later steps of the M$^{pro}$-mediated proteolysis reaction. Deciphering the underlying, general principles that connect peptide substrate sequence and conformation to M$^{pro}$-mediated cleavage efficiency remains a significant but important challenge for future structural and computational studies of coronavirus main protease enzymes (*Kenward et al., 2024*; *Cesar Ramos de Jesus et al., 2024*; *Anirudhan et al., 2021*; *Wang et al., 2021*; *Narwal et al., 2023*; *Brewitz et al., 2022*).

Overall, our work defines the structural basis for understanding recognition and cleavage of the human tRNA methyltransferase TRMT1 by the SARS-CoV-2 main protease and directly shows that TRMT1 cleavage leads to inactivation of its tRNA m2,2G methyltransferase activity. *Zhang et al., 2024*, provide complementary evidence that TRMT1 is proteolyzed and inactivated during SARS-CoV-2 infection, which would be predicted to affect host and viral translation and increase cellular sensitivity to redox stress. TRMT1 cleavage may therefore have important implications for understanding the ability of SARS-CoV-2 to hijack host protein synthesis and impact cellular phenotypes and COVID-19 pathogenesis. Additionally, our TRMT1-M$^{pro}$ structure highlights a distinct substrate binding mode that reveals the S3′ pocket on the surface of the main protease. SARS-CoV-2 M$^{pro}$ is vital for viral propagation and is thus a major target for antiviral drug development (*Burkhardt et al., 2023*; *Cwilichowska et al., 2022*; *Katre et al., 2022*; *Steuten et al., 2021*; *Narayanan et al., 2022*). Consideration of the often-hidden S3′ pocket on the M$^{pro}$ surface could aid in the design of potent and specific SARS-CoV-2 M$^{pro}$ inhibitors. Currently, the vast majority of inhibitors target the S2 and S1 sites of M$^{pro}$, with peptidyl inhibitors designed with glutamine and leucine analogs for the P2 and P1 positions, respectively (*Agost-Beltrán et al., 2022*). Our structural results suggest the flexible S3′ pocket could be utilized in future therapeutic design, and together with our kinetic analysis adds to the growing understanding of M$^{pro}$ substrate recognition and cleavage efficiency that is critical for potent and specific targeting of M$^{pro}$ to prevent or treat infection.

# Materials and methods

### Key resources table

| Reagent type (species) or resource | Designation | Source or reference | Identifiers | Additional information |
|---|---|---|---|---|
| Recombinant DNA reagent (plasmid) | pLVX-EF1alpha-SARS-CoV-2-nsp5-2xStrep-IRES-Puro | Addgene | RRID:Addgene_141370 | |
| Recombinant DNA reagent (plasmid) | pLVX-EF1alpha-SARS-CoV-2-nsp5-C145A-2xStrep-IRES-Puro | Addgene | RRID:Addgene_141371 | |
| Recombinant DNA reagent (plasmid) | pETARA-M$^{pro}$-WT | This paper | | Mugridge Lab plasmid, *E. coli* expression vector for wild-type M$^{pro}$ with a GST tag |
| Recombinant DNA reagent (plasmid) | pETARA-M$^{pro}$-C145A | This paper | | Mugridge Lab plasmid, *E. coli* expression vector for M$^{pro}$ C145A variant with a GST tag |
| Recombinant DNA reagent (plasmid) | pETARA-M$^{pro}$-M49A | This paper | | Mugridge Lab plasmid, *E. coli* expression vector for M$^{pro}$ M49A variant with a GST tag |
| Recombinant DNA reagent (plasmid) | pETARA-M$^{pro}$-N142A | This paper | | Mugridge Lab plasmid, *E. coli* expression vector for M$^{pro}$ N142A variant with a GST tag |
| Recombinant DNA reagent (plasmid) | pETARA-M$^{pro}$-Q189A | This paper | | Mugridge Lab plasmid, *E. coli* expression vector for M$^{pro}$ Q189A variant with a GST tag |
| Recombinant DNA reagent (plasmid) | TRMT1-pET22b(+)–18del-WT | GenScript | | *E. coli* expression plasmid to generate recombinant human TRMT1 WT |
| Peptide | TRMT1(526–536) unlabeled Peptide | Peptide 2.0 | | EPRLQANFTIR |
| Peptide | TRMT1(526–536) labeled Peptide | Peptide 2.0 | | MCA-EPRLQANFTIR-K(Dnp)K |
| Peptide | nsp4/5 labeled Peptide | Peptide 2.0 | | MCA-SAVLQSGFRKM-K(Dnp)K |
| Peptide | nsp8/9 labeled Peptide | Peptide 2.0 | | MCA-AVKLQNNELSP- K(Dnp)K |
| Software, algorithm | Origin software | OriginLab Corporation | RRID:SCR_014212; Versions 2021 and 2023b | |
| Software, algorithm | GraphPad Prism software | GraphPad | RRID:SCR_002798; Version 10.0.03 | |
| Cell line (*Homo sapiens*) | 293T | ATCC | ATCC: CRL-3216 | |
| Antibody | Anti-TRMT1 460–659 | Invitrogen | PA5-96585 | Western blot primary (1:2000) |

*Continued on next page*

*Continued*

| Reagent type (species) or resource | Designation | Source or reference | Identifiers | Additional information |
|---|---|---|---|---|
| Antibody | Anti-TRMT1 609–659 (Rabbit polyclonal) | Bethyl Laboratories | A304-205A | Western blot primary (1:2000) |
| Antibody | Anti-GAPDH (Rabbit polyclonal) | Invitrogen | PA-1987 | Western blot primary (1:100,000) |
| Antibody | Goat anti-Rabbit IgG (H+L) Secondary Antibody, HRP | Invitrogen | A16096 | Western blot secondary (1:10,000) |
| Commercial assay or kit | Clarity Western ECL Substrate | Bio-Rad Laboratories | 1705060 | HRP substrate for western blot |
| Recombinant DNA reagent | tRNA Phe$_{GAA}$, sequence with T7 promoter (*Homo sapiens*) | IDT | | TAATACGACTCACTATAGCCG AAATAGCTCAGTTGGG AGAGCGTTAGACTGAAGA TCTAAAGGTCCCT GGTTCGATCCCGGGTTTCGGCA |
| Commercial assay or kit | RNA Clean & Concentrator kits | Zymo Research | R1016 | For purification of tRNA product |
| Software, algorithm | ImageJ software | *Schneider et al., 2012* | Version 1.54d | For image analysis of EMSA gels |
| Other | Spark microplate reader | Tecan | | For plate-based activity assay |
| Other | FluorChem R imager | Protein Simple | | For imaging western blots |
| Other | Quantulus Scintillation Counter | Revvity | | For measuring $^{14}$C CPM |
| Software, algorithm | Coot software | *Emsley et al., 2010* | Version 0.8.9.1 | Structure building |
| Software, algorithm | PHENIX software | *Liebschner et al., 2019* | Version v.1.17.1–3600 | Structure refinement |

## Cloning, protein expression, and purification

M$^{pro}$ constructs used for biochemical and kinetic studies were subcloned into a pETARA expression vector containing N-terminal GST and C-terminal His-tags using Gibson assembly. Human codon optimized WT M$^{pro}$ was obtained from Addgene (catalog #141370). M$^{pro}$ single-point mutants (M49A, N142A, Q189A) were introduced using site-directed mutagenesis by whole plasmid PCR. Each construct contained the M$^{pro}$ autocleavage sequence (AVLQ) after the N-terminal GST tag to allow for self-cleavage and a native M$^{pro}$ N-terminus. Constructs were transformed into *E. coli* Rosetta(DE3) pLysS cells and plated on LB agar plates with 50 µg/mL ampicillin. Overnight seed cultures were grown from single colonies in LB media with 50 µg/mL ampicillin, at 37°C with shaking at 200 rpm. A 1:100 dilution of seed culture to 1 L LB media was prepared and grown at 37°C with shaking at 200 rpm. After an OD$_{600}$ of 0.6 was reached, cells were induced with 1 mM IPTG and incubated overnight at 18°C with shaking. Cultures were centrifuged at 7500 × *g* and the supernatant was discarded. Harvested cells were sonicated in lysis buffer (25 mM Tris, 300 mM NaCl, 10 mM imidazole pH 8.0), centrifuged at 14,500 × *g* for 45 min, and recovered, clarified lysate was loaded onto 1.5 mL (bed volume) equilibrated Thermo Scientific HisPur Ni-NTA Resin for 30 min with gentle mixing. Resin was added to a standard gravity column and washed with two 25 mL washes of lysis buffer and two 25 mL washes with lysis buffer+25 mM imidazole, and eluted with 10 mL lysis buffer+250 mM imidazole. Eluate was concentrated to ~2 mL using a using 10 kDa MWCO centrifugal concentrator and applied to a Cytiva HiLoad 16/600 Superdex 200 pg column for size exclusion chromatography. M$^{pro}$-containing fractions were concentrated to between 10 and 25 mg/mL, flash-frozen with liquid nitrogen, and stored at –70°C in 50 mM Tris, 1 mM EDTA, 2 mM DTT, pH 7.3.

Catalytically inactive M$^{pro}$ C145A mutants used for protein crystallography experiments were prepared by subcloning M$^{pro}$ C145A from Addgene (catalog #141371) into a pET28a expression vector containing an N-terminal His-tag and tobacco etch virus (TEV) protease cleavage site (ENLYFQGS). The M$^{pro}$ C145A construct was transformed and expressed in *E. coli* as described above. Cultures were centrifuged at 7500 × *g* and the supernatant was discarded. Harvested pellets were sonicated in lysis buffer (25 mM Tris, 300 mM NaCl, pH 8.0) and clarified the lysate by centrifugation at 14,500 × *g*. Recovered lysate was loaded onto 3.75 mL (bed volume) equilibrated Thermo Scientific HisPur Ni-NTA Resin for 30 min with gentle mixing. Resin was added to a standard gravity column and washed with

two 25 mL washes of lysis buffer and two 25 mL washes with lysis buffer+25 mM imidazole, and eluted with lysis buffer+300 mM imidazole. Eluate was incubated overnight at room temperature with recombinant purified TEV protease at a 1:50 molar ratio to cleave the N-terminal His-tag. An Ni-NTA backpass was performed and the flow-through was concentrated, buffer exchanged into 25 mM Tris, 25 mM NaCl, pH 7.5 using a using 10 kDa MWCO centrifugal concentrator, and applied to a Cytiva HiTrap 5 mL Q HP column. Protein was eluted with a gradient of 25 mM to 1 M NaCl and M$^{pro}$-containing fractions were pooled, concentrated, and applied to a Cytiva HiLoad 16/600 Superdex 200 pg size exclusion chromatography column and eluted with an isocratic gradient using 25 mM Tris, 25 mM NaCl, pH 7.5. The protein-containing fractions were concentrated to approximately 26 mg/mL, flash-frozen with 10% glycerol in liquid nitrogen, and stored at –70°C.

The human TRMT1 gene sequence (NM_001136035.4) with a 1–18 amino acid N-terminal deletion (to remove the mitochondrial targeting peptide) was synthesized and subcloned by GenScript into pET22b(+) with an N-terminal pelB leader sequence and a C-terminal His-tag. The TRMT1 construct was transformed into *E. coli* Rosetta(DE3)pLysS cells as described above, and induced using 0.2 mM IPTG. Harvested cells were sonicated in lysis buffer (20 mM Tris-HCl, pH 8.5, 500 mM NaCl, 5 mM imidazole, 10% glycerol, 5 mM BME, cOmplete EDTA-free Protease Inhibitor Cocktail), clarified at 14,500 × *g*, and recovered lysate was loaded onto a pre-equilibrated Cytiva 5 mL HisTrap column. The loaded HisTrap column washed with lysis buffer+20 mM imidazole, and protein was eluted using a gradient elution with lysis buffer+250 mM imidazole. Fractions containing TRMT1 were pooled, concentrated, and loaded onto a Cytiva HiLoad 16/600 Superdex 200 pg size exclusion chromatography column and eluted with an isocratic gradient using 20 mM Tris pH 8.5, 500 mM NaCl, 2 mM DTT. The protein-containing fractions were concentrated to approximately 30 mg/mL, flash-frozen in liquid nitrogen, and stored at –70°C.

## Crystallography and structure determination

Purified M$^{pro}$ C145A was incubated at room temperature for 1 hr with TRMT1 (526–536) peptide (EPRLQANFTIR, synthetic peptide obtained from Peptide 2.0) in 25 mM Tris, 25 mM NaCl, pH 7.4 at a 1:3 molar ratio with the final concentrations of 7 mg/mL M$^{pro}$ C145A and 619 μM TRMT1 (526–536). The M$^{pro}$ and TRMT1 solution was mixed 1:1 with well solutions consisting of 20–21.5% PEG 3350 and 100 mM NaCl in 24-well hanging drop VDX plates with a final drop volume of 1 μL. Initial crystals were harvested, crushed, and seeded into new hanging drops, with components as listed above, using cat whiskers. Single crystals were harvested and flash-frozen in liquid nitrogen with 20% glycerol in well solution as a cryoprotectant. Diffraction data were collected at the National Synchrotron Light Source II (NSLS II) Highly Automated Macromolecular Crystallography (AMX) (*Schneider et al., 2022*) beamline 17-ID-1 at the Brookhaven National Laboratory on an Eiger 9M Pixel detector at 100 K and a wavelength of 0.920219 Å. Diffraction data were indexed, integrated, and scaled using XDS. The M$^{pro}$-TRMT1 structure was solved in space group P2$_1$2$_1$2$_1$ using the Phaser package in the CCP4 suite and a modified PDB 7MGS as search model. After initial rounds of refinement in PHENIX to model M$^{pro}$ residues, the TRMT1 peptide residues 526–534 were manually built into the $F_o$-$F_c$ map using COOT. Subsequent rounds of automated refinement and water placement using PHENIX and manual adjustments including modeling sodium ions and glycerol molecules in COOT were used to obtain the final structure.

## Peptide cleavage kinetic assays

Fluorogenic assays measuring kinetic parameters for cleavage of peptide substrates were carried out in triplicate in Corning Low Volume 384-well Black Flat Bottom Polystyrene NBS Microplates following a similar procedure as *Lee et al., 2020*. Peptide cleavage reactions were carried out at 50 nM M$^{pro}$ enzyme in 50 mM Tris pH 7.3, 1 mM EDTA, 2 mM DTT, and 20% DMSO, with quenched fluorescent peptide substrates ranging in concentration from 0.097 to 100 μM. Synthetic peptide substrates were obtained from Peptide 2.0: TRMT1 (MCA-EPRLQANFTIR-K(Dnp)K), nsp4/5 (MCA-SAVLQSGFRKM-K(Dnp)K), and nsp8/9 (MCA-AVKLQNNELSP- K(Dnp)K) where MCA = 7-methoxycoumarin-4-acetic acid and Dnp = dinitrophenyl. Using a Tecan Spark microplate reader, the fluorescence intensity was monitored every 10 s over a 3 min time course, with excitation at 320 nm and emission at 405 nm. A calibration curve of MCA-AVLQ product fluorescence intensities from 12 to 0.006 μM was measured and used to generate an RFU to μM conversion factor. A correction for the inner filter effect (IFE$_{corr}$)

was determined using the formula IFE$_{corr}$ = [fluorescence$_{MCA\_product+peptide}$ − fluorescence$_{peptide}$]/fluorescence$_{MCA}$ where fluorescence$_{MCA\_product+peptide}$ = the fluorescence of MCA-AVLQ mixed with quenched peptide substrate, fluorescence$_{peptide}$ = the fluorescence of the peptide alone, fluorescence$_{MCA}$ = fluorescence of MCA-AVLQ product alone (*Figure 5—source data 2*; *Liu et al., 1999*). Plots of initial rate (µM/min) vs peptide substrate concentration were fit to the Michaelis-Menten equation to determine $V_{max}$ and $K_M$ kinetic parameters for M$^{pro}$-mediated peptide cleavage using Origin 2021 software.

## Cell lines

HEK293T cells used in this study were obtained from ATCC with confirmed identity by STR profiling and no detection of mycoplasma, bacterial, or fungal contamination.

## Mammalian cell culture

Freshly passaged HEK293T cells were incubated at 37°C with 5% CO$_2$ in growth media (DMEM media with 10% FBS and penicillin [100 U/mL]/streptomycin (100 µg/mL)). Growth media was replaced every 3 days, and cells were trypsinized and passaged approximately every 7 days, when cells reached 80–90% confluency. To passage, growth media was aspirated, and cells were gently washed with prewarmed PBS. PBS was then aspirated and a 1:2.5 mixture of 0.25% Trypsin in HBSS with 0.2 g/L EDTA to PBS was added to adherent cells and incubated at 37°C for 2 min to disaggregate. A 3× dilution with growth media was used to inactivate trypsin. The cells were gently mixed to attain a homogenous suspension and diluted 1:20 with growth media and added into a new cell culture plate. For lysis, cells were trypsinized as described above, and the resulting cell suspension was spun down at 1250 rpm for 3 min. Cell pellets were washed in ice-cold PBS three times, and spun at 1250 rpm for 3 min, aspirating PBS in between each wash. Lysis was performed on a PBS suspension of cells by performing 3 consecutive freeze/thaw cycles by flash-freezing in liquid nitrogen and thawing at 37°C. Lysate was collected after centrifugation at 13,000 rpm for 10 min and stored at –70°C.

## Proteolysis assay

TRMT1 proteolysis reactions with recombinant TRMT1 isolated from *E. coli* were performed with 10 µM WT or C145A M$^{pro}$ in 20 mM HEPES, 105 mM NaCl, 20% glycerol, pH 7.0. Recombinant FL TRMT1 from *E. coli* was diluted sevenfold. Reaction was incubated at 37°C and timepoints were quenched by adding to SDS-PAGE sample buffer (50 mM Tris, pH 6.8, 1.0% [wt/vol] SDS, 10% [vol/vol] glycerol, 0.1% [wt/vol] bromophenol blue, 0.1 M DTT) and boiled for 5 min. TRMT1 levels and fragment sizes at different reaction timepoints were assessed by western blot.

TRMT1 proteolysis assays using endogenous human TRMT1 from HEK293T lysate were carried out using cell lysate preincubated with 1 mM phenylmethylsulfonyl fluoride prior to reaction to prevent proteolysis by mammalian-specific proteases. M$^{pro}$ proteolysis reactions were performed with 10 µM WT or C145A M$^{pro}$ in 25 mM Tris, 25 mM NaCl, 20% glycerol, pH 7.5. Reaction was incubated at 37°C and timepoints were quenched by adding to SDS-PAGE sample buffer (50 mM Tris, pH 6.8, 1.0% [wt/vol] SDS, 10% [vol/vol] glycerol, 0.1% [wt/vol] bromophenol blue, 0.1 M DTT) and boiled for 5 min. TRMT1 levels and fragment sizes at different reaction timepoints were assessed by western blot.

## Western blot

Quenched samples from the proteolysis assays were loaded (10 µL) onto Bio-Rad TGX 4–15% polyacrylamide gels and run for 30 min at 180 V. Gels were blotted onto PVDF membranes using a Bio-Rad Trans blot Turbo for 7 min at 2.5 A. The blot was incubated in blocking solution (5% non-fat milk in 1× Tris-Buffer Saline with 0.1% Tween) at room temperature for 1 hr. All antibodies were diluted in blocking solution. TRMT1-specific antibodies corresponding to amino acid regions 460–659 (Invitrogen Rabbit Anti-TRMT1 [ref: PA5-96585]) and 609–659 (Bethyl Laboratories Rabbit Anti-TRMT1 [ref: A304-205A]) and housekeeping protein GAPDH (Invitrogen Rabbit Anti-GAPDH [ref: PA-1987]) were utilized for primary antibody staining at 1:2,000 dilution for TRMT1 antibodies and 1:100,000 for GAPDH antibodies, and were stained by overnight incubation at 4°C. Extensive washing of blot with 1× Tris-buffered saline with 0.1% Tween were performed after primary antibody staining. Invitrogen Goat anti-Rabbit IgG (H+L) Secondary Antibody, HRP (ref: A16096) was used for secondary staining at a dilution of 1:10,000 for 1 hr at room temperature. Extensive washing of blot with 1× Tris-buffered saline with 0.1% Tween were performed after secondary antibody staining. Clarity Western ECL

Substrate (ref: 1705060) was added to blot and incubated at room temperature for 5 min. Western blots were visualized on a Protein Simple FluorChem R imager.

## In vitro transcription of tRNA substrate

A double-stranded template of tRNA$^{phe}$ (*Homo sapiens* Phe-GAA, sequence with T7 promoter: TAATACGACTCACTATAGCCGAAATAGCTCAGTTGGGAGAGCGTTAGACTGAAGATCTAAAGGtCCCTGGTTCGATCCCGGGTTTCGGCA) was used at a concentration of 150 nM in the transcription mixture of 0.05% Triton X-100, 5 mM DTT, 5 mM rNTPs, 0.1 mg/mL RNA polymerase, 2 U/mL TIPP, 40 mM Tris-HCl, pH 7.5, 50 mM MgCl$_2$, and 2 mM spermidine. The reaction was incubated at 37°C for 3 hr, then treated with DNase RQ1 and incubated at 37°C for 1 hr. The reaction was quenched with 50 mM EDTA and 50% formamide and heated for 5 min at 95°C. The quenched material was loaded onto a large-scale 8% gel PAGE denaturing UREA gel and run at 50 mA for ~2.5 hr. Using a handheld UV lamp and a silica plate, the tRNA band was excised, shred, and nutated overnight at room temperature in 0.6 M NaOAc, pH 6.0, 1 mM EDTA, and 0.01% SDS. The solution was filtered and extracted using phenol:chloroform:IAA, 25:24:1, pH 6.6, centrifuging for 5 min at 1500 × *g* and collecting the aqueous phase, then precipitating with 80% ethanol overnight at –20°C. The tRNA pellet was air-dried at room temperature and brought up in molecular grade water. tRNA was then re-folded by heating at 80°C for 2 min, 60°C for 2 min, and then adding 10 mM MgCl$_2$ and cooling on ice for 30 min. The final material was flash-frozen at ~150 μM and stored at –80°C.

## $^{14}$C radiolabel-based methyltransferase activity assays

Overnight M$^{pro}$ cleavage reactions were prepared with 0.5 μM TRMT1 in 50 mM Tris pH 8.0, 50 mM MgCl$_2$, 100 μg/μL BSA, 2 mM DTT and added to 20 μM M$^{pro}$ WT, 20 μM M$^{pro}$ C145A, or no protease. Timepoints at 0 and 18 hr were taken for each protease condition and assessed for reaction completion by western blot. The completed M$^{pro}$ WT, M$^{pro}$ C154A, or no protease (mock) cleavage reactions were used directly in methyltransferase activity assays with 0.1 μM final concentration of TRMT1, 5 μM tRNA$^{phe}$, and 30 μM $^{14}$C SAM (4.2 mCi/mmol) in 50 mM Tris pH 8.0, 50 mM MgCl$_2$, 100 μg/μL BSA, 2 mM DTT. Reactions were incubated at 37°C and 5 μL timepoints were taken at 0, 0.5, 2, and 4 hr and processed using Zymo RNA Clean & Concentrator kits to isolate tRNA from the reaction mixture. tRNA was eluted in 15 μL water and added to 10 mL of Ultima Gold Scintillation Fluid (ref: 6013329). Counts per minute (CPM) were measured using Quantulus Scintillation Counter CH3 channel (lower level 0, upper level 156, 1 min count time). A standard curve using known concentrations and specific activity of $^{14}$C-SAM was produced to convert CPM to concentration of labeled product. Graphs were generated using Origin2023b and fit using a Michaelis-Menten equation.

## EMSA for detecting tRNA binding

Overnight M$^{pro}$ cleavage reactions were prepared with 14 μM TRMT1 in 20 mM Tris-HCl, pH 8.5, 300 mM NaCl, 2 mM DTT, 20% glycerol and added to 140 μM M$^{pro}$ WT, 140 μM M$^{pro}$ C145A, or no protease. Timepoints at 0 and 18 hr were taken for each protease condition and assessed for reaction completion by western blot. The completed M$^{pro}$ WT, M$^{pro}$ C154A, or no protease (mock) cleavage reactions were used directly to generate a dilution series of TRMT1 and each dilution was mixed with tRNA$^{phe}$ (final TRMT1 concentrations ranged from 7.78 to 0.13 μM; final tRNA$^{Phe}$ concentration was 0.2 μM). The TRMT1-tRNA binding reactions were incubated for 1 hr at room temperature, 8 μL of each reaction was loaded onto a 5% TBE non-denaturing gel, and run at 50–60 V for approximately 2.5 hr at 4°C. Each gel was stained with 1× SYBR Gold in 0.5× TBE for 30 min at room temperature and then imaged using a Protein Simple FluorChemQ imager. Band intensities for bound and unbound tRNA were quantified using ImageJ software (*Schneider et al., 2012*) and used to calculate fraction bound values at each TRMT1 concentration. Graphs were generated using Origin2023b and fit using a Hill plot.

## MD simulation

Extensive MD simulations in explicit water have been carried out to investigate M$^{pro}$-nsp4/5 and M$^{pro}$-TRMT1 complexes starting from their corresponding crystal structures (PDB id: 7MGS [*MacDonald et al., 2021*] and 8D35), respectively. In original crystal structures, the catalytic Cys145 has been mutated to Ala, M$^{pro}$-nsp4/5 is in the monomer form, while the M$^{pro}$-TRMT1 complex is in the dimer

form. Since the active form of $M^{pro}$ should be dimer (**Zhang et al., 2020b**), the structure of the $M^{pro}$-nsp4/5 (PDB id: 7MGS) was superimposed to the $M^{pro}$-TRMT1 (PDB id: 8D35) to model the dimer form of $M^{pro}$-nsp4/5, and the catalytic Cys145 in both complexes has been modeled using 'swapaa' command in ChimeraX (**Pettersen et al., 2021**). The original crystal water molecules and ions within 5 Å of the protein-substrate complex are kept. PDB2PQR (**Jurrus et al., 2018**) was employed to add hydrogen atoms to both complexes at pH 7. Subsequently, the His41 was manually changed to Hip41 (positively charged) by adding an additional hydrogen atom to the imidazole ring in ChimeraX, and Cys145 was manually changed to Cym (negatively charged) by removing the hydrogen atom on the thiol group in ChimeraX. Each system was neutralized by adding counterions for $Na^+$ and $Cl^-$, and solvated in a rectangular water box of TIP3P water molecules with 12 Å buffer.

For each prepared simulation system, 3000 steps of steepest descent plus 2000 steps of conjugate minimization was performed with harmonic restraints using a force constant of 20 kcal/mol/Å² applied to all heavy atoms coming from the crystal structure, and then the whole system was minimized without restraints by 7000 steps of steepest descent and 3000 steps of conjugate gradient minimization. After minimization, three independent equilibration and MD replicas were carried out for 100 ns with the same starting structural configuration but different initial velocities. The equilibration was conducted in five steps: (a) 50 ps constant volume ensemble (NVT) MD simulation with 10 kcal/mol/ Å² restraints on all heavy atoms from the crystal, and the whole system was heated from 10 to 300 K gradually. (b) 50 ps isothermal isobaric ensemble (NPT) MD with 10 kcal/mol/Å² restraints on all heavy atoms from the crystal at 300 K. (c) 200 ps NPT MD with 5 kcal/mol/Å² restraints on all heavy atoms from the crystal at 300 K. (d) 200 ps NPT MD with 2 kcal/mol/Å² restraints on all heavy atoms from the crystal at 300 K. (e) 200 ps NPT MD with 1 kcal/mol/Å² restraints on all heavy atoms from the crystal at 300 K. Finally, production MD simulations were carried out for 100 ns at a constant temperature of 300 K and a constant pressure of 1 atm. Langevin thermostat and Berendsen barostat methods were employed to maintain the temperature and pressure, respectively. The atomic coordinates of the complexes were saved every 1 ps to obtain the trajectories for analysis.

All MD simulations were conducted with Amber 20 (**Case, 2020**) package using Amber FF 14SB force field (**Maier et al., 2015**). AmberTools (**Case, 2020**) was utilized for preparing topology and coordinate files for the simulated systems cutoff of 10 Å was set for calculating van der Waals interactions, and the particle mesh Ewald (**Darden et al., 1993**) method with a cutoff of 10 Å was employed to treat electrostatic interactions. The SHAKE algorithm (**Andersen, 1983**) was used to constrain covalent bonds to allow the integration time step of 2 fs.

## Evolutionary analysis of mammalian TRMT1 orthologs

For the mammalian-wide sequence analyses, the TRMT1 amino acid alignment was retrieved with OrthoMaM (Orthologous Mammalian Markers) v10c (**Scornavacca et al., 2019**), using human TRMT1 (ENSG00000104907) as query. The cleavage site was located and the sequence logo of the 526–536 region was generated using WebLogo3 (https://weblogo.threeplusone.com/).

For the primate phylogenetic analyses, TRMT1 orthologous sequences were retrieved using the DGINN pipeline with the human TRMT1 CCDS12293 as query (**Picard et al., 2020**). TRMT1 sequences from additional primate species were retrieved using NCBI Blastn (**Figure 6—source data 1**).

For the rodent phylogenetic analyses, the rodentia orthologous protein and mRNA reference sequences to the human TRMT1 (ENSG00000104907) gene were collected from OrthoMaM and ncbi HomoloGene (https://www.ncbi.nlm.nih.gov/homologene; **Figure 6—source data 1**). Because most rodent species lack a reference CDS, we used BLOSUM62 matrix implemented in GeneWise from EMBL-EBI tools (**Birney et al., 2004**; **Madeira et al., 2019**), to identify the TRMT1 ORFs. Codon alignments of primate and rodent TRMT1s were performed using WebPrank (**Löytynoja and Goldman, 2010**) with default settings for primates (trust insertions + F, gap rate = 0.05, gap length = 5, K=2) and with a gap rate of 0.1 for rodents. Positive selection analyses on the codon alignments were performed with HYPHY/Datamonkey (**Kosakovsky Pond et al., 2020**; **Weaver et al., 2018**), using two branch-site models, BUSTED (version 3.1)(**Murrell et al., 2015**) and aBSREL (version 2.2) (**Smith et al., 2015**), and two site-specific models, MEME (version 2.1.2) (**Murrell et al., 2012**) and FUBAR (version 2.2) (**Murrell et al., 2013**).

To identify missense polymorphisms or variants in TRMT1 at minor allele frequency above 0.005 in human population, we mined the dbSNP database https://www.ncbi.nlm.nih.gov/snp/.

## Materials availability

All materials, including plasmids generated in this study, are available from the authors upon request.

## Acknowledgements

We would like to thank A Soares at Brookhaven National Laboratory, National Synchrotron Light Source II and the Schmitz lab at University of Delaware for assistance with X-ray data collection and X Liu at UCSF for advice on structure refinement. We are grateful to JSM's cat Percival for the generous donation of whiskers used in protein crystal seeding experiments. We thank A Cimarelli and the LP2L team members at CIRI for feedback. This work was supported by the US National Institutes of Health, National Institute of General Medical Sciences, under awards R35 GM143000 to JSM, R35 GM127040 to YZ, T32 GM133395 CBI fellowship to AD, and P20 GM104316 and S10 OD026896A that funded key instrumentation used in this study. The content is solely the responsibility of the authors and does not necessarily represent the official views of the National Institutes of Health. The work was further supported by the CNRS and the French Agence Nationale de la Recherche (ANR), under grant ANR-20-CE15-0020-01 (project 'BATantiVIR') and by a grant from the French Research Agency on HIV and Emerging Infectious Diseases ANRS/MIE (#ECTZ118944) to LE Computational resources were provided by NYU-ITS. This research used resources at AMX beamline 17-ID-1 of the National Synchrotron Light Source II, a US Department of Energy (DOE) Office of Science User Facility operated for the DOE Office of Science by Brookhaven National Laboratory under Contract No. DE-SC0012704. The Center for BioMolecular Structure (CBMS) is primarily supported by the National Institutes of Health, National Institute of General Medical Sciences (NIGMS) through a Center Core P30 Grant (P30GM133893), and by the DOE Office of Biological and Environmental Research (KP1607011).

## Additional information

### Funding

| Funder | Grant reference number | Author |
| --- | --- | --- |
| National Institutes of Health | R35 GM143000 | Jeffrey S Mugridge |
| National Institutes of Health | R35 GM127040 | Yingkai Zhang |
| National Institutes of Health | T32 GM133395 | Angel D'Oliviera |
| National Institutes of Health | P20 GM104316 | Jeffrey S Mugridge |
| National Institutes of Health | S10 OD026896A | Jeffrey S Mugridge |
| Agence Nationale de la Recherche | ANR-20-CE15-0020-01 | Lucie Etienne |
| French Research Agency on HIV and Emerging Infectious Diseases ANRS/MIE | ECTZ118944 | Lucie Etienne |

The funders had no role in study design, data collection and interpretation, or the decision to submit the work for publication.

### Author contributions

Angel D'Oliviera, Conceptualization, Data curation, Formal analysis, Validation, Investigation, Visualization, Methodology, Writing – original draft, Writing – review and editing; Xuhang Dai, Saba Mottaghinia, Data curation, Formal analysis, Validation, Investigation, Visualization, Methodology, Writing – review and editing; Sophie Olson, Formal analysis, Investigation, Methodology; Evan P Geissler, Investigation; Lucie Etienne, Yingkai Zhang, Resources, Data curation, Formal analysis,

Supervision, Funding acquisition, Validation, Investigation, Visualization, Methodology, Writing – review and editing; Jeffrey S Mugridge, Conceptualization, Resources, Data curation, Formal analysis, Supervision, Funding acquisition, Validation, Investigation, Visualization, Methodology, Writing – original draft, Project administration, Writing – review and editing

### Author ORCIDs
Angel D'Oliviera (ID) https://orcid.org/0000-0002-5944-9609
Xuhang Dai (ID) https://orcid.org/0009-0007-6712-8737
Saba Mottaghinia (ID) https://orcid.org/0000-0002-9409-3302
Lucie Etienne (ID) https://orcid.org/0000-0002-8585-7534
Yingkai Zhang (ID) https://orcid.org/0000-0002-4984-3354
Jeffrey S Mugridge (ID) https://orcid.org/0000-0002-1553-3008

Reviewer #1 (Public review): https://doi.org/10.7554/eLife.91168.3.sa1
Reviewer #2 (Public review): https://doi.org/10.7554/eLife.91168.3.sa2
Reviewer #3 (Public review): https://doi.org/10.7554/eLife.91168.3.sa3
Author response https://doi.org/10.7554/eLife.91168.3.sa4

## Additional files

### Supplementary files
MDAR checklist

### Data availability
Coordinates and structure factors were deposited in the Protein Data Bank with accession code 9DW6 (note 9DW6 is a re-refinement with very minor changes to our original deposition, 8D35, and supersedes this entry). All other data generated and analyzed in this study are available as source data and supporting files associated with this manuscript, or for additional data used in sequence analysis can be found at https://doi.org/10.6084/m9.figshare.22004474 and https://doi.org/10.6084/m9.figshare.22004492.

The following datasets were generated:

| Author(s) | Year | Dataset title | Dataset URL | Database and Identifier |
|---|---|---|---|---|
| D'Oliviera A, Mugridge JS | 2024 | Crystal structure of SARS-CoV-2 main protease (Mpro) C145A mutant in complex with peptide from human tRNA methyltransferase TRMT1 | https://www.rcsb.org/structure/9DW6 | RCSB Protein Data Bank, 9DW6 |
| Etienne L | 2023 | Primate TRMT1 codon alignment | https://doi.org/10.6084/m9.figshare.22004474 | figshare, 10.6084/m9.figshare.22004474 |
| Etienne L | 2023 | Rodent TRMT1 codon alignment | https://doi.org/10.6084/m9.figshare.22004492 | figshare, 10.6084/m9.figshare.22004492 |

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
