## [Editor Report · eLife Assessment]

This manuscript provides **important** structural insights into the recognition and degradation of the host tRNA methyltransferase TRMT1 by SARS-CoV-2 protease nsp5 (Mpro). The data provide **compelling** support for the main conclusions of the authors. These results will be of interest to researchers studying structures, substrate recognition and specificity of viral proteases and their action on cellular targets.

---

## [Referee Report · Reviewer #1 (Public review)]

D'Oliviera et al. have demonstrated cleavage of human TRMT1 by the SARS-CoV-2 main protease in vitro. Following, they solved the structure of Mpro (Nsp5)-C145A bound to TRMT1 substrate peptide, revealing binding conformation distinct from most viral substrates. Overall, this work enhances our understanding of substrate specificity for a key drug target of CoV2. The paper is well-written and the data is clearly presented. It complements the companion article by demonstrating interaction between Mpro and TRMT1, as well as TRMT1 cleavage under isolated conditions in vitro. They show that cleaved TRMT1 has reduced tRNA binding affinity, linking a functional consequence to TRMT1 cleavage by MPro. Importantly, the revelation for flexible substrate binding of Nsp5 is fundamental for understanding Nsp5 as a drug target. Trmt1 cleavage assays by Mpro revealed similar kinetics for TRMT1 cleavage as compared to nsp8/9 viral polyprotein cleavage site. They purify TRMT1-Q350K, in which there is a mutation in the predicted cleavage consensus sequence, and confirm that it is resistant to cleavage by recombinant Mpro. I am unable to comment critically on the structural analyses as it is outside of my expertise. Overall, I think that these findings are important for confirming TRMT1 as a substrate of Mpro, defining substrate binding and cleavage parameters for an important drug target of SARS-CoV-2, and may be of interest to researchers studying RNA modifications.

---

## [Referee Report · Reviewer #2 (Public review)]

Summary:

The manuscript 'Recognition and Cleavage of Human tRNA Methyltransferase TRMT1 by the SARS-CoV-2 Main Protease' from Angel D'Oliviera et al., uncovers that TRMT1 can be cleaved by SARS-CoV-2 main protease (Mpro) and defines the structural basis of TRMT1 recognition by Mpro. They use both recombinant TRMT1 and Mpro as well as endogenous TRMT1 from HEK293T cell lysates to convincingly show cleavage of TRMT1 by the SARS-CoV-2 protease. Using in vitro assays, the authors demonstrate that TRMT1 cleavage by Mpro blocks its enzymatic activity leading to hypomodification of RNA. To understand how Mpro recognizes TRMT1, they solved a co-crystal structure of Mpro bound to a peptide derived from the predicted cleavage site of TRMT1. This structure revealed important protein-protein interfaces and highlights the importance of the conserved Q530 for cleavage by Mpro. They then compare their structure with previous X-ray crystal structures of Mpro bound to substrate peptides derived from the viral polyprotein and propose the concept of two distinct binding conformations to Mpro: P3´-out and P3´-in conformations (here P3´ stands for the third residue downstream of the cleavage site). It remains unknown what is the physiological role of these two binding conformations on Mpro function, but the authors established that Mpro has dramatically different cleavage efficiencies for three distinct substrates. In an effort to rationalize this observation, a series of mutations in Mpro's active site and the substrate peptide were tested but unexpectedly had no significant impact on cleavage efficiency. While molecular dynamic simulations further confirmed the propensity of certain substrates to adopt the P3´-out or P3´-in conformation, it did not provide additional insights into the dramatic differences in cleavage efficiencies between substrates. This led the authors to propose that the discrimination of Mpro for preferred substrates might occur at a later stage of catalysis after binding of the peptide. Overall, this work will be of interest to biologists studying proteases and substrate recognition by enzymes and RNA modifications as well as help efforts to target Mpro with peptide-like drugs.

Strengths:

• The authors' statements are well supported by their data, and they used relevant controls when needed. Indeed, they used the Mpro C145A inactive variant to unambiguously show that the TRMT1 cleavage detected in vitro is solely due to Mpro's activity. Moreover, they used two distinct polyclonal antibodies to probe TRMT1 cleavage.

• They demonstrate the impact of TRMT1 cleavage on RNA modification by quantifying both its activity and binding to RNA.

• Their 1.9 Å crystal structure is of high quality and increases the confidence in the reported protein-protein contacts seen between TRMT1-derived peptide and Mpro.

• Their extensive in vitro kinetic assay was performed in ideal conditions although it is sometimes unclear how many replicates were performed.

• They convincingly show how Mpro cleavage is conserved among most but not all mammalian TRMT1 bringing an interesting evolutionary perspective on virus-host interactions.

• The authors test multiple hypotheses to rationalize the preference of Mpro for certain substrates.

• While this reviewer is not able to comment on the rigor of the MD simulations, the interpretations made by the authors seem reasonable and convincing.

• The concept of two binding conformations (P3´-out or P3´-in) for the substrate in the active site of Mpro is significant and can guide drug design.

Weaknesses:

• The two polyclonal antibodies used by the authors seem to have strong non-specific binding to proteins other than TRMT1 but did not impact the author's conclusions or statements. This is a limitation of the commercially available antibodies for TRMT1.

• Despite the reasonable efforts of the authors, it remains unknown why Mpro shows higher cleavage efficiency for the nsp4/5 sequence compared to TRMT1 or nsp8/9 sequences. This is a challenging problem that will take substantially more effort by several labs to decipher mechanistically.

• The peptide cleavage kinetic assay used by the authors relies on a peptide labelled with a fluorophore (MCA) on the N-terminus and a quencher (Dpn) on the C-terminus. This design allows high-throughput measurements compatible with plate readers and is a robust and convenient tool. Nevertheless, the authors did not control for the impact of the labels (MCA and Dpn) on the activity of Mpro. While in most cases the introduced fluorophore/quencher do not impact activity, sometimes it can.

• An unanswered question not addressed by the authors is if the peptides undergo conformational changes upon Mpro binding or if they are pre-organized to adopt the P3´-out and P3´-in conformations. This might require substantially more work outside the scope of this immediate article.

---

## [Referee Report · Reviewer #3 (Public review)]

Summary:

In this manuscript, the authors have used a combination of enzymatic, crystallographic, and in silico approaches to provide compelling evidence for substrate selectivity of SARS-CoV-2 Mpro for human TRMT1.

Strengths:

In my opinion, the authors came close to achieving their intended aim of demonstrating the structural and biochemical basis of Mpro catalysis and cleavage of human TRMT1 protein. The revised version of the manuscript has addressed most of the questions I had posed in my earlier review.

Weaknesses:

Although several new hypotheses are generated from the Mpro structural data, the manuscript falls a bit short of testing them in functional assays, which would have solidified the conclusions the authors have drawn.

---

## [Author Response]

The following is the authors’ response to the current reviews.

**Public Reviews:**

**Reviewer #1 (Public review):**
D'Oliviera et al. have demonstrated cleavage of human TRMT1 by the SARS-CoV-2 main protease in vitro. Following, they solved the structure of Mpro (Nsp5)-C145A bound to TRMT1 substrate peptide, revealing binding conformation distinct from most viral substrates. Overall, this work enhances our understanding of substrate specificity for a key drug target of CoV2. The paper is well-written and the data is clearly presented. It complements the companion article by demonstrating interaction between Mpro and TRMT1, as well as TRMT1 cleavage under isolated conditions in vitro. They show that cleaved TRMT1 has reduced tRNA binding affinity, linking a functional consequence to TRMT1 cleavage by MPro. Importantly, the revelation for flexible substrate binding of Nsp5 is fundamental for understanding Nsp5 as a drug target. Trmt1 cleavage assays by Mpro revealed similar kinetics for TRMT1 cleavage as compared to nsp8/9 viral polyprotein cleavage site. They purify TRMT1-Q350K, in which there is a mutation in the predicted cleavage consensus sequence, and confirm that it is resistant to cleavage by recombinant Mpro. I am unable to comment critically on the structural analyses as it is outside of my expertise. Overall, I think that these findings are important for confirming TRMT1 as a substrate of Mpro, defining substrate binding and cleavage parameters for an important drug target of SARS-CoV-2, and may be of interest to researchers studying RNA modifications.

We thank the reviewer for their positive assessment and summary of our work in this paper!

**Reviewer #2 (Public review):**
Summary:The manuscript 'Recognition and Cleavage of Human tRNA Methyltransferase TRMT1 by the SARS-CoV-2 Main Protease' from Angel D'Oliviera et al., uncovers that TRMT1 can be cleaved by SARS-CoV-2 main protease (Mpro) and defines the structural basis of TRMT1 recognition by Mpro. They use both recombinant TRMT1 and Mpro as well as endogenous TRMT1 from HEK293T cell lysates to convincingly show cleavage of TRMT1 by the SARS-CoV-2 protease. Using in vitro assays, the authors demonstrate that TRMT1 cleavage by Mpro blocks its enzymatic activity leading to hypomodification of RNA. To understand how Mpro recognizes TRMT1, they solved a co-crystal structure of Mpro bound to a peptide derived from the predicted cleavage site of TRMT1. This structure revealed important protein-protein interfaces and highlights the importance of the conserved Q530 for cleavage by Mpro. They then compare their structure with previous X-ray crystal structures of Mpro bound to substrate peptides derived from the viral polyprotein and propose the concept of two distinct binding conformations to Mpro: P3´-out and P3´-in conformations (here P3´ stands for the third residue downstream of the cleavage site). It remains unknown what is the physiological role of these two binding conformations on Mpro function, but the authors established that Mpro has dramatically different cleavage efficiencies for three distinct substrates. In an effort to rationalize this observation, a series of mutations in Mpro's active site and the substrate peptide were tested but unexpectedly had no significant impact on cleavage efficiency. While molecular dynamic simulations further confirmed the propensity of certain substrates to adopt the P3´-out or P3´-in conformation, it did not provide additional insights into the dramatic differences in cleavage efficiencies between substrates. This led the authors to propose that the discrimination of Mpro for preferred substrates might occur at a later stage of catalysis after binding of the peptide. Overall, this work will be of interest to biologists studying proteases and substrate recognition by enzymes and RNA modifications as well as help efforts to target Mpro with peptide-like drugs.

We thank the reviewer for this thorough and accurate summary of our work in this manuscript.

Strengths:• The authors' statements are well supported by their data, and they used relevant controls when needed. Indeed, they used the Mpro C145A inactive variant to unambiguously show that the TRMT1 cleavage detected in vitro is solely due to Mpro's activity. Moreover, they used two distinct polyclonal antibodies to probe TRMT1 cleavage.• They demonstrate the impact of TRMT1 cleavage on RNA modification by quantifying both its activity and binding to RNA.• Their 1.9 Å crystal structure is of high quality and increases the confidence in the reported protein-protein contacts seen between TRMT1-derived peptide and Mpro.• Their extensive in vitro kinetic assay was performed in ideal conditions although it is sometimes unclear how many replicates were performed.• They convincingly show how Mpro cleavage is conserved among most but not all mammalian TRMT1 bringing an interesting evolutionary perspective on virus-host interactions.• The authors test multiple hypotheses to rationalize the preference of Mpro for certain substrates.• While this reviewer is not able to comment on the rigor of the MD simulations, the interpretations made by the authors seem reasonable and convincing.• The concept of two binding conformations (P3´-out or P3´-in) for the substrate in the active site of Mpro is significant and can guide drug design.

We thank the reviewer for these positive assessments of manuscript strengths!

Weaknesses:• The two polyclonal antibodies used by the authors seem to have strong non-specific binding to proteins other than TRMT1 but did not impact the author's conclusions or statements. This is a limitation of the commercially available antibodies for TRMT1.

Yes, there are some levels of non-specific binding for all of the TRMT1 antibodies we have tested (this limitation of commercially available TRMT1 antibodies is also observed and noted by Zhang et al), but we agree that this does not impact the overall conclusions and that by using multiple different antibodies to show the same effects, we can have high confidence in the Western blot analysis and interpretation.

• Despite the reasonable efforts of the authors, it remains unknown why Mpro shows higher cleavage efficiency for the nsp4/5 sequence compared to TRMT1 or nsp8/9 sequences. This is a challenging problem that will take substantially more effort by several labs to decipher mechanistically.

True! To our knowledge and despite significant past efforts of many research groups studying similar coronavirus proteases (e.g. SARS-CoV-1 Mpro) a clear understanding of the detailed mechanistic relationship between cleavage sequence and cleavage kinetics remains mostly undefined. This is a great and important problem for mechanistic and computational groups with deep interests in proteases to tackle in the future! To highlight these and similar open questions, we have added a short paragraph to the Discussion section (second from the last paragraph).

• The peptide cleavage kinetic assay used by the authors relies on a peptide labelled with a fluorophore (MCA) on the N-terminus and a quencher (Dpn) on the C-terminus. This design allows high-throughput measurements compatible with plate readers and is a robust and convenient tool. Nevertheless, the authors did not control for the impact of the labels (MCA and Dpn) on the activity of Mpro. While in most cases the introduced fluorophore/quencher do not impact activity, sometimes it can.

Yes, we agree that it is possible the MCA and Dnp labels could have effects on the measured cleavage rates. These fluorophore/quencher peptide cleavage assays are the standard assays used by many labs in the protease field to study diverse proteases and diverse cleavage targets. When other labs have compared cleavage kinetic parameters measured with fluorophore/quencher-based peptide cleavage assays versus HPLC-based peptide cleavage assays, these are often found to be quite similar (e.g. Lee, J., Worrall, L.J., Vuckovic, M. et al. Crystallographic structure of wild-type SARS-CoV-2 main protease acyl-enzyme intermediate with physiological C-terminal autoprocessing site. Nat Commun 11, 5877 (2020). https://doi.org/10.1038/s41467-020-19662-4), although there are also examples where differences arise. In any case, we agree there could be some effects on the cleavage kinetics introduced by the fluorophore and/or quencher groups. However, our main focus in this paper is to show how a sequence in the human tRNA-modifying enzyme TRMT1 is cleaved by Mpro (and in this revision we have also added new data to show the functional effects of cleavage on TRMT1 activity); it will take significant future work to fully dissect the detailed relationships between peptide sequence, including the quantitative effects of fluorophore/quencher labels, and protease-directed cleavage kinetics. Based on our work in this paper and many past studies of similar proteases, understanding how peptide sequence or conformation relates to cleavage efficiency is a longer-term and very challenging problem that we view as beyond the scope of this work. We have added a brief section elaborating on this in the Discussion.

• An unanswered question not addressed by the authors is if the peptides undergo conformational changes upon Mpro binding or if they are pre-organized to adopt the P3´-out and P3´-in conformations. This might require substantially more work outside the scope of this immediate article.

We agree this is unanswered; we considered additional MD experiments to address this, but ultimately decided that since both of these sequences are cleaved in the context of much larger polypeptides (FL TRMT1 or the viral polypeptide), any simple analysis to assess the possibility of pre-organization and relate this preferred binding conformation to cleavage kinetics would be difficult to interpret in a biologically meaningful way. We think this and similar questions about how pre-organization of peptides or amino acid sequences in the polypeptides might influence protease binding and cleavage activity are interesting and important future questions for protease-focused groups in this field.

**Reviewer #3 (Public review):**
Summary:In this manuscript, the authors have used a combination of enzymatic, crystallographic, and in silico approaches to provide compelling evidence for substrate selectivity of SARS-CoV-2 Mpro for human TRMT1.Strengths:In my opinion, the authors came close to achieving their intended aim of demonstrating the structural and biochemical basis of Mpro catalysis and cleavage of human TRMT1 protein. The revised version of the manuscript has addressed most of the questions I had posed in my earlier review.

We thank the reviewer for their positive assessment of this work, and we are glad to hear the manuscript revisions were helpful in addressing the first round of reviews and questions.

Weaknesses:Although several new hypotheses are generated from the Mpro structural data, the manuscript falls a bit short of testing them in functional assays, which would have solidified the conclusions the authors have drawn.

Toward showing some of the functional effects of TRMT1 cleavage, in this revised version of the manuscript we have added new data and a new results section (‘Cleavage of TRMT1 results in complete loss of tRNA m2,2G modification activity and reduced tRNA binding in vitro’) showing that cleavage of TRMT1 results in reduced tRNA binding to TRMT1 (Figure 2D) and the complete loss of TRMT1-mediated tRNA modification activity in vitro (Figure 2C). This complements the in-cell data presented by Zhang et al showing that cleavage of TRMT1 in SARS-CoV-2 infected human cells results in the reduction of m2,2G modification levels. We think these data are a strong addition to this paper that broadens the impacts of our reported results more directly into the RNA modifications field.

In terms of showing the further, downstream biological effects of TRMT1 cleavage and/or the specific impacts of TRMT1 cleavage on SARS-CoV-2 propagation and replication, while we agree further functional assays could absolutely heighten the overall impact, we view the main focus of our paper as showing how TRMT1 is recognized and cleaved by Mpro at the structural level and characterizing the biochemistry of the TRMT1-Mpro interaction and the effects of cleavage on TRMT1 tRNA-modifying activity. Zhang et al present some cellular data suggesting that loss of TRMT1 and/or TRMT1 cleavage during infection is actually detrimental to SARS-CoV-2 replication and infectivity. However, a full understanding of how TRMT1-mediated m2,2G modification of tRNA impacts viral translation, whether TRMT1 plays other roles during the viral life cycle, or whether TRMT1 cleavage (even if not important for viral fitness) contributes to cellular phenotypes during infection, will take a significant amount of future cell biology and virology work to unravel. Indeed, our understanding is that characterizing some of the endogenous cleavage targets for the HIV protease and determining the downstream biological effects and impacts on HIV infection took well over a decade. We hope that the biochemical and structural characterization of the Mpro-TRMT1 interaction presented in our paper will provide the necessary fundamental groundwork and impetus for future virology and cellular biochemistry studies to further investigate the biological roles of TRMT1 cleavage by SARS-CoV-2 Mpro.

---

The following is the authors’ response to the original reviews.

**eLife Assessment:**
This manuscript provides important structural insights into the recognition and degradation of the host tRNA methyltransferase by SARS-CoV-2 protease nsp5 (Mpro). The data convincingly support the main conclusions of the paper. These results will be of interest to researchers studying structures and substrate recognition and specificity of viral proteases.

We thank the eLife editors and reviewers for handling this manuscript and the overall positive assessment of our work.

In this revised version of the manuscript we have included significant, new experimental data with recombinant purified, catalytically active TRMT1 that directly shows cleavage of TRMT1 reduces its tRNA binding affinity (by gel shift assays) and results in the complete loss of tRNA modifying activity in vitro (by radiolabel-based methyltransferase assays). Because these added experiments provide new information about how Mpro-mediated cleavage specifically impacts TRMT1 tRNA binding and m2,2G modification activity, and thus new information about the functional effects of loss of the TRMT1 Zn finger domain, we would strongly suggest adding that “this work may be of interest to researchers studying RNA modifications”, or a similar phrase, in the eLife assessment.

Please find below our point-by-point response to each of the reviewer comments, which outlines additional changes to the manuscript.

**Public Reviews:**

**Reviewer #1 (Public Review):**
D'Oliviera et al. have demonstrated cleavage of human TRMT1 by the SARS-CoV-2 main protease in vitro. Following this, they solved the structure of Mpro-C145A bound to TRMT1 substrate peptide, revealing binding conformation distinct from most viral substrates. Overall, this work enhances our understanding of substrate specificity for a key drug target of CoV2. The paper is well-written and the data is clearly presented. It complements the companion article by demonstrating the interaction between Mpro and TRMT1 and TRMT1 cleavage under isolated conditions in vitro. Importantly, the revelation of flexible substrate binding of Nsp5 is fundamental for understanding Nsp5 as a drug target. Trmt1 cleavage assays revealed similar kinetics for TRMT1 cleavage as compared to the nsp8/9 viral polyprotein cleavage site, however, it would have been more rigorous for the authors to independently reproduce the kinetics reported for nsp8/9 using their specific experimental conditions. The finding that murine TRMT1 lacks a conserved consensus sequence is interesting, but is not experimentally tested here and is reported elsewhere. I am unable to comment critically on the structural analyses as it is outside of my expertise. Overall, I think that these findings are important for confirming TRMT1 as a substrate of Mpro and defining substrate binding and cleavage parameters for an important drug target of SARS-CoV-2.

We thank the reviewer for their positive assessment and summary of our work in this paper!

We absolutely agree that comparing to nsp8/9 cleavage kinetics measured in our own hands would be more rigorous here, and we have carried out these measurements in triplicate under the same conditions as were used to measure all the other peptide cleavage kinetics in this manuscript. Figures 5A & B (as well as Table S3 and Dataset S2) have been updated with our new nsp8/9 kinetic data (kcat = 0.019 +/- 0.002 s-1 and KM = 40 +/- 7.5 µM). As expected, our newly measured nsp8/9 kinetic parameters are very similar to those that we had previously cited from MacDonald et al (kcat = 0.013 +/- 0.001 s-1, KM = 36 +/- 6.0 µM), and show that Mpro-mediated TRMT1 peptide cleavage has similar proteolysis kinetics to the nsp8/9 viral polypeptide cleavage site.

We have also purified full-length human TRMT1 Q530K, which is the key change in the cleavage consensus sequence that likely makes murine TRMT1 resistant to Mpro-mediated cleavage. In in vitro cleavage assays we find that indeed TRMT1 Q530K is entirely resistant to cleavage by recombinant Mpro and we have added this data to the manuscript in Figure 6D. These findings are consistent with previously cited data from Lu et al, which suggest mouse and hamster TRMT1 are not cleaved in HEK293T cells expressing Mpro.

With the addition of the TRMT1 Q530K mutant data, we decided to move the evolutionary analysis together with this kinetic data to a new section in the Results. We think these additions and changes make the paper stronger and clearer, and thank the reviewer for these suggestions!

**Reviewer #2 (Public Review):**
Summary:The manuscript 'Recognition and Cleavage of Human tRNA Methyltransferase TRMT1 by the SARS-CoV-2 Main Protease' from Angel D'Oliviera et al., uncovers that TRMT1 can be cleaved by SARS-CoV-2 main protease (Mpro) and defines the structural basis of TRMT1 recognition by Mpro. They use both recombinant TRMT1 and Mpro as well as endogenous TRMT1 from HEK293T cell lysates to convincingly show cleavage of TRMT1 by the SARS-CoV-2 protease. To understand how Mpro recognizes TRMT1, they solved a co-crystal structure of Mpro bound to a peptide derived from the predicted cleavage site of TRMT1. This structure revealed important protein-protein interfaces and highlights the importance of the conserved Q530 for cleavage by Mpro. They then compared their structure with previous X-ray crystal structures of Mpro bound to substrate peptides derived from the viral polyprotein and proposed the concept of two distinct binding conformations to Mpro: P3´-out and P3´-in conformations (here P3´ stands for the third residue downstream of the cleavage site). It remains unknown what is the physiological role of these two binding conformations on Mpro function, but the authors established that Mpro has dramatically different cleavage efficiencies for three distinct substrates. In an effort to rationalize this observation, a series of mutations in Mpro's active site and the substrate peptide were tested but unexpectedly had no significant impact on cleavage efficiency. While molecular dynamic simulations further confirmed the propensity of certain substrates to adopt the P3´-out or P3´-in conformation, they did not provide additional insights into the dramatic differences in cleavage efficiencies between substrates. This led the authors to propose that the discrimination of Mpro for preferred substrates might occur at a later stage of catalysis after binding of the peptide. Overall, this work will be of interest to biologists studying proteases and substrate recognition by enzymes as well as help efforts to target Mpro with peptide-like drugs.

We thank the reviewer for this thorough and accurate summary of our work in this manuscript.

Strengths:• The authors' statements are well supported by their data, and they used relevant controls when needed. Indeed, they used the Mpro C145A inactive variant to unambiguously show that the TRMT1 cleavage detected in vitro is solely due to Mpro's activity. Moreover, they used two distinct polyclonal antibodies to probe TRMT1 cleavage.• Their 1.9 Å crystal structure is of high quality and increases the confidence in the reported protein-protein contacts seen between TRMT1-derived peptide and Mpro.• Their extensive in vitro kinetic assay was performed in ideal conditions although it is unclear how many replicates were performed.• The authors test multiple hypotheses to rationalize the preference of Mpro for certain substrates.• While this reviewer is not able to comment on the rigor of the MD simulations, the interpretations made by the authors seem reasonable and convincing.• The concept of two binding conformations (P3´-out or P3´-in) for the substrate in the active site of Mpro is significant and can guide drug design.

We thank the reviewer for these positive assessments of manuscript strengths!

Weaknesses:• While the authors convincingly show that TRMT1 is cleaved by Mpro, the exact cleavage site was never confirmed experimentally. It is most likely that the predicted site is the main cleavage site as proposed by the authors (region 527-534). Nevertheless, in Fig 1C (first lane from the right) there are two bands clearly observed for the cleavage product containing the MT Domain. If the predicted site was the only cleavage site recognized by Mpro, then a single band for the MT domain would be expected. This observation suggests that there might be two cleavage sites for Mpro in TRMT1. Indeed, residues RFQANP (550-555) in TRMT1 might be a secondary weaker cleavage site for Mpro, which would explain the two observed bands in Fig 1C. A mass spectrometry analysis of the cleaved products would clarify this.

We agree with the reviewer that based on the originally presented data it is possible there could be an additional Mpro-targeted cleavage site in TRMT1 beyond the 527-534 region that we validated through peptide cleavage assays of the TRMT1 526-536 peptide. Because it may be difficult to unambiguously identify and differentiate other putative cleavage sites that are nearby to 527-534 (e.g. the suggested possibility of 550-555) by mass spectrometry, we instead carried out additional in vitro cleavage assays with purified FL TRMT1 Q530K. Mutation of the invariant P1 Gln residue in the cleavage sequence is expected to prevent cleavage at this site, and allow us to probe whether there are other sites in TRMT1 that can be cleaved by Mpro (and if so, more straightforwardly identify them by mass spectrometry). We compared cleavage of purified WT FL TRMT1 and FL TRMT1 Q530K with recombinant Mpro in in vitro cleavage assays and found that TRMT1 Q530K is not cleaved by Mpro over the course of a 2h cleavage reaction. In these experiments, we also saw clear cleavage of WT FL TRMT1 over the course of 2h into only a single detectable band. Together, both of these pieces of data strongly suggest that the 527-534 region is the only Mpro-targeted cleavage site in TRMT1 (if there was an additional cleavage site, we should have seen some amount of cleavage in the Q530K mutant, but we do not). Overall, we feel that the updated WT and Q530K experiments clearly demonstrate that there is only one Mpro-mediated cleavage site in human TRMT1, which also is consistent with experiments in Zhang et al showing that Q530N mutations also block TRMT1 cleavage by co-expressed Mpro in human cells.

The updated WT and Q530K cleavage assays have been added to the manuscript in Figure 6D.

• A control is missing in Fig 1D. Since the authors use western blots to show the gradual degradation of endogenous TRMT1, a control with a protein that does not change in abundance over the course of the measurement is important. This is required to show that the differences in intensity of TRMT1 by western blotting are not due to loading differences etc.

Yes, we agree this is an important control and have repeated these experiments and blotted for TRMT1 and GAPDH as a loading control. The updated Western blots are now shown in Figure 2B, and show the same result as the older data.

• The two polyclonal antibodies used by the authors seem to have strong non-specific binding to proteins other than TRMT1 but did not impact the author's conclusions. This is a limitation of the commercially available antibodies for TRMT1, and unless the authors select a new monoclonal antibody specific to TRMT1 (costly and lengthy process), this limitation seems out of their control.

Yes, there are some levels of non-specific binding for all of the TRMT1 antibodies we have tested (this limitation of commercially available TRMT1 antibodies is also observed and noted by Zhang et al), but we agree that this does not impact the overall conclusions and that by using multiple different antibodies to show the same effects, we can have high confidence in the Western blot analysis and interpretation.

• The recombinantly purified TRMT1 seems to have some non-negligible impurities (extra bands in Fig 1C). This does not impact the conclusions of the authors but might be relevant to readers interested in working with TRMT1 for biochemical, structural, or other purposes.

Yes, our initial isolations of recombinant TRMT1 for the first version of this paper produced smaller amounts of TRMT1 with some impurities; we agree that these do not impact the conclusions of the cleavage experiments. However, since our first submission, we have optimized our purification protocols for TRMT1 and are now able to obtain larger quantities of higher purity recombinant human TRMT1 from bacterial cells and we have used this material for the TRMT1 activity and tRNA binding assays added in this revision; we have also included updates to the expression and purification section for recombinant TRMT1. We hope that these improvements will be helpful to readers interested in working on TRMT1.

• Despite the reasonable efforts of the authors, it remains unknown why Mpro shows higher cleavage efficiency for the nsp4/5 sequence compared to TRMT1 or nsp8/9 sequences.

True! To our knowledge and despite significant past efforts of many research groups studying similar coronavirus proteases (e.g. SARS-CoV-1 Mpro) a clear understanding of the detailed mechanistic relationship between cleavage sequence and cleavage kinetics remains mostly undefined. This is a great and important problem for mechanistic and computational groups with deep interests in proteases to tackle in the future! To highlight these and similar open questions, we have added a short paragraph to the Discussion section (second from the last paragraph).

• The peptide cleavage kinetic assay used by the authors relies on a peptide labelled with a fluorophore (MCA) on the N-terminus and a quencher (Dpn) on the C-terminus. This design allows high-throughput measurements compatible with plate readers and is a robust and convenient tool. Nevertheless, the authors did not control for the impact of the labels (MCA and Dpn) on the activity of Mpro. It is possible that the differences in cleavage efficiencies between peptides are due to unexpected conformational changes in the peptide upon labelling. Moreover, the TRMT1 peptide has an E at the N-terminus and an R at the C-terminus (while the nsp4/5 peptide has an S and M, respectively). It is possible that these two terminal residues form a salt bridge in the TRMT1 peptide that might constrain the conformation of the peptide and thus reduce its accessibility and cleavage by Mpro. Enzymatic assays in the absence of labels and MD simulations with the bona fide peptides (including the labels) used in the kinetic measurements are needed to prove that the cleavage efficiencies are not biased by the fluorescence assay.

These fluorophore/quencher peptide cleavage assays are the standard assays used by many labs in the protease field to study diverse proteases and diverse cleavage targets. When other labs have compared cleavage kinetic parameters measured with fluorophore/quencher-based peptide cleavage assays versus HPLC-based peptide cleavage assays, these are often found to be quite similar (e.g. Lee, J., Worrall, L.J., Vuckovic, M. et al. Crystallographic structure of wild-type SARS-CoV-2 main protease acyl-enzyme intermediate with physiological C-terminal autoprocessing site. Nat Commun 11, 5877 (2020). https://doi.org/10.1038/s41467-020-19662-4), although there are also examples where differences arise. In any case, we agree there could be some effects on the cleavage kinetics introduced by the fluorophore and/or quencher groups or sequence-specific conformational preferences of the peptides. However, because our main focus in this paper is to show how a sequence in the human tRNA-modifying enzyme TRMT1 is cleaved by Mpro (and in this revision we have also added new data to show the functional effects of cleavage on TRMT1 activity), and the broad focus of our lab is understanding the mechanisms controlling the function and activity of RNA-modifying enzymes, we will leave it to other labs focused more specifically on protease biochemistry to fully dissect the detailed relationships between peptide sequence and conformation to protease-directed cleavage kinetics. As discussed above, based on our work in this paper and many past studies of similar proteases, understanding how sequence relates to cleavage efficiency is a longer-term and very challenging problem that we view as beyond the scope of this work. As noted above, we have added a brief section explaining this in the Discussion.

• The authors used A431S variant in TRMT1-derived peptide to disrupt the P3´-in conformation. While this reviewer agrees with the rationale behind A431S design, it is important to confirm experimentally that the mutation disrupted the P3´-in conformation in favor of the P3´-out conformer. The authors could use their MD simulations to determine if the TRMT1 A431S variant favors the P3´-out conformation.

Thank you for this suggestion; we agree and have carried out the suggested MD simulations with TRMT1 A531S peptides bound to Mpro. Surprisingly, these simulations suggest that the A531S peptide can still readily adopt the P3’-in conformation by orienting the Ser sidechain in a different way as compared to its positioning in the Mpro-nsp4/5 structure. Since this somewhat changes our interpretation of the results of the A531S kinetic experiments, we have rewritten this section of the manuscript by: (a) removing the ‘TRMT1 mutations predicted to alter peptide binding conformation have little effect on cleavage kinetics’ section in the Results, (b) instead adding several sentences talking about the A531S mutation to the previous section of the results, and including this mutation as another example of how mutations to either Mpro or TRMT1 residues that might be expected to impact cleavage kinetics do not in fact affect cleavage rates, and finally (c) adding the new MD simulation results to the A531S kinetic data in Figure S5 in the Supporting Information. We thank the reviewer for suggesting this important follow-up simulation!

• An unanswered question not addressed by the authors is if the peptides undergo conformational changes upon Mpro binding or if they are pre-organized to adopt the P3´-out and P3´-in conformations.

We agree this is unanswered; we considered additional MD experiments to address this, but ultimately decided that since both of these sequences are cleaved in the context of much larger polypeptides (FL TRMT1 or the viral polypeptide), any simple analysis to assess the possibility of pre-organization and relate this preferred binding conformation to cleavage kinetics would be difficult to interpret in a biologically meaningful way. We think this and similar questions about how pre-organization of peptides or amino acid sequences in the polypeptides might influence protease binding and cleavage activity are interesting and important future questions for protease-focused groups in this field.

• While the authors describe at great length the hydrogen bonds involved in the substrate recognition by Mpro, they occluded to highlight important stacking interactions in this interface. For instance, Phe533 from TRMT1 stacks with Met49 while L529 from TRMT1 packs against His41 of Mpro. Both hydrogen bonding and stacking interactions seem important for TRMT1-derived peptide recognition by Mpro.

Thank you for these suggestions toward additional structural analysis. We have added a short description of L529 packing in the S2 pocket to the main text and Figure S3B. We have also added a short description of F533 packing in the S3’ pocket to the main text and Figure S3C.

**Reviewer #3 (Public Review):**
Summary:In this manuscript, the authors have used a combination of enzymatic, crystallographic, and in silico approaches to provide compelling evidence for substrate selectivity of SARS-CoV-2 Mpro for human TRMT1.Strengths:In my opinion, the authors came close to achieving their intended aim of demonstrating the structural and biochemical basis of Mpro catalysis and cleavage of human TRMT1 protein. The combination of orthogonal approaches is highly commendable.

We thank the reviewer for their positive assessment of this work!

Weaknesses:It would have been of high scientific impact if the consequences of TRMT1 cleavage by Mpro on cellular metabolism were provided. Furthermore, assays to investigate the effect of inhibition of this Mpro activity on SARS-CoV-2 propagation and infection would have been extremely useful in providing insights into host- SARS-CoV-2 interactions.

Toward showing some of the consequences of TRMT1 cleavage, in this revised version of the manuscript we have added new data and a new results section (‘Cleavage of TRMT1 results in complete loss of tRNA m2,2G modification activity and reduced tRNA binding in vitro’) showing that cleavage of TRMT1 results in reduced tRNA binding to TRMT1 (Figure 2D) and the complete loss of TRMT1-mediated tRNA modification activity in vitro (Figure 2C). This complements the in-cell data presented by Zhang et al showing that cleavage of TRMT1 in SARS-CoV-2 infected human cells results in the reduction of m2,2G modification levels. We think these data are a strong addition to this paper that broadens the impacts of our reported results more directly into the RNA modifications field.

In terms of showing the further, downstream biological effects of TRMT1 cleavage and/or the specific impacts of TRMT1 cleavage on SARS-CoV-2 propagation and replication, while we agree this would absolutely heighten the overall impact, we view the main focus of our paper as showing how TRMT1 is recognized and cleaved by Mpro at the structural level and characterizing the biochemistry of the TRMT1-Mpro interaction and the effects of cleavage on TRMT1 tRNA-modifying activity. Zhang et al present some cellular data suggesting that loss of TRMT1 and/or TRMT1 cleavage during infection is actually detrimental to SARS-CoV-2 replication and infectivity. However, a full understanding of how TRMT1-mediated m2,2G modification of tRNA impacts viral translation, whether TRMT1 plays other roles during the viral life cycle, or whether TRMT1 cleavage (even if not important for viral fitness) contributes to cellular phenotypes during infection, will take a significant amount of future cell biology and virology work to unravel. Indeed, our understanding is that characterizing some of the endogenous cleavage targets for the HIV protease and determining the downstream biological effects and impacts on HIV infection took well over a decade. We hope that the biochemical and structural characterization of the Mpro-TRMT1 interaction presented in our paper will provide the necessary fundamental groundwork and impetus for future virology and cellular biochemistry studies to further investigate the biological roles of TRMT1 cleavage by SARS-CoV-2 Mpro.

Recommendations for the authors:Reviewer #1 (Recommendations For The Authors):Please list Mpro alias Nsp5 in the Abstract and Introduction, as this is the nomenclature used in the companion article.

OK, we have made these changes.

Reviewer #2 (Recommendations For The Authors):In addition to the points mentioned in the public review, this reviewer encourages the authors to address the following points:• Citation 14 is important for this work since the authors used multiple structures from that earlier study for comparison. Citation 14 seems outdated since it refers to a preprint that has been published since then in Nat Comm. The authors should cite the peer-reviewed work https://pubmed.ncbi.nlm.nih.gov/35729165/

Thank you, we have updated this reference.

• The description of the hydrogen bonds is tedious to read. The authors could instead classify them into two groups. Hydrogen bonds between main chain backbones or hydrogen bonds between side chains. For instance, they mention the contact between Mpro Glu166-TRMT1 Arg528. This can lead to confusion that a salt bridge is formed while these two residues interact only via their main chain backbones. Indeed, the side chain of R528 is exposed to the solvent.

OK, we have taken this suggestion and tried to simplify and clarify this portion of the text (along with the accompanying structure Figure 3 showing key hydrogen bonds; see below).

• For Figure 2, please label the residues of the peptide with the TRMT1 numbering. This will help the reader to follow the text while looking at the figure.

OK we have added the TRMT1 numbering to what is now Figure 3A, and labeled key TRMT1 residues in Figures 3B, C, and D.

• Fig 2B is important but crowded. The authors could use two panels to show two different views of this interface.

Thank you for this suggestion, we have split B (now C and D in Figure 3) into two panels, rotated 90 degrees from one another, with each view showing a different subset of TRMT1-Mpro interactions. These updated panels are less crowded, and will hopefully be much clearer to readers.

• For increased clarity, the authors could color P3´-out in orange and P3´-in teal in Fig 3D.

OK, we have made this change.

• Please proofread the method section. There should be a space between values and their units. For example, 20mM HEPES should be 20 mM HEPES.

Thank you, we have corrected these formatting errors in the methods section of the revised version of the manuscript.

• The authors did not identify the mechanism for the higher efficiency of nsp4/5 cleavage despite testing several mutants and MD simulations. Did the author consider changes in the network of water molecules that might be identified in the MD simulations?

We did look at the positioning of waters in nsp4/5 vs nsp8/9 vs TRMT1 MD simulations. In the nsp4/5 simulation we do see a slightly higher density of water molecules positioned at approximately reasonable attack angles for substrate hydrolysis. If we consider water molecules with an attack angle on the scissile amide of 82 – 96 degrees and an attack distance of 4 Å or closer, the probabilities for these conditions in the simulations are: nsp4/5 – 19%, nsp8/9 – 9%, TRMT1 – 6%. More water positioned at reasonable attack positions for nsp4/5 might be consistent with its higher cleavage efficiency, but: (a) these are relatively small differences in water positioning across these 3 Mpro-substrate simulations that would not be enough to clearly explain the large differences in observed kinetics, and (b) hydrolysis happens in the later steps of the catalytic cycle, so to accurately capture this we would likely need to simulate reaction intermediates formed after initial attack of the active site Cys.

We very much appreciate the reviewer’s enthusiasm in pushing us to understand the mechanistic basis for Mpro-directed cleavage efficiencies, and we would have absolutely loved to figure this out! (As it appears to be a long-standing question in the field!) But as discussed above and in the manuscript, we think that it will take a detailed dissection of different steps in the catalytic cycle to understand where and how this selectivity arises. We will leave it to research groups focused more exclusively on the details of protease biochemistry and simulations of reactive intermediates to take up these significant and long-term challenges!

• In the PDB deposition, Y154 from chain B should be fixed.• In the PDB deposition, some added glycerols seem to conflict. Although this is not important for the biological work discussed in this study, the authors should check if glycerol 403 in chain A and 402, 403 in chain B are properly modeled. Does the density justify placing a glycerol there?• In the PDB deposition, there are over 51 RSRZ outliers. The authors should double-check if they cannot fix them with additional refinements. While such outliers in poorly defined linkers are understandable, this is unexpected for well-defined regions in the map.

We have made a number of updates to our PDB deposition to address the above three points. (1) We have reexamined and tweaked the loop region at Y154 chain B; this region of the structure has relatively poorly defined electron density, but we now have a model where Y154 is no longer a Ramachandran outlier. The PDB model is now free of any Ramachandran outliers. (2) We have reexamined each of the modeled glycerol molecules and removed one of these (GOL 402), which had a weaker fit to the electron density. The remaining two glycerols appear to be well-modeled (omit maps leaving out each glycerol show strong Fo-Fc density that clearly looks like a glycerol in shape, adding each glycerol back into the model decreases Rwork and Rfree, and the refined 2Fo-Fc map fits well to the modeled glycerols). (3) We agree there are a large number of RSRZ outliers in this structure. We have reexamined many of these, and come to the same conclusion as for our original deposition: that most of these result from residues where there is clear enough density for placing the backbone into the map, but very poor density for the sidechain. Modeling different sidechain positions for the RSRZ outliers we reexamined did not appreciably improve the model fit or change their RSRZ outlier status. For example, Y154 in chains A and B remain some of the worst RSRZ outliers; while the density for these loop regions is generally not very good, it is clear that the backbone atoms of Y154 can be modeled into the structure, but there is very very weak density for the sidechain. We tried modeling alternative and/or multiple sidechain conformations for Y154, but this did not significantly reduce the size of the RSRZ outlier. In short, while we could remove some of these residues or truncate the sidechain where the sidechain density is very poor to lower the total number of RSRZ outliers, we think the best model is one where we leave these residues built into the structure and accept the higher number of RSRZ outliers. Importantly, none of the significant RSRZ outliers are key residues of biological interest that would affect our interpretation of the structure and/or TRMT1-Mpro biochemistry.

We have deposited a new, re-refined PDB model (9DW6) that incorporates these changes and supersedes our old PDB entry (8D35). We have updated the manuscript with the new PDB ID. We thank the reviewer for these suggestions that improved the overall structural model.

**Reviewer #3 (Recommendations For The Authors):**
The crystal structure entry in the PDB should mention the Cys-to-Ala substitution in Mpro.

Thank you, we have made this change

Fig 2A and 2B: Can the authors highlight the Gln520-Ala531 peptide bind with a different color, please? It gets lost in panel B.

Yes, we have made significant revisions to what is now Figure 3, and have highlighted the scissile peptide bond atoms in orange in each of these panels. Thank you for this suggestion, we agree it helps readers to orient themselves within the structure.

"Importantly, the identified Mpro-targeted residues in human TRMT1 are conserved in the human population (i.e. no missense polymorphisms), showing that human TRMT1 can be recognized and cleaved by SARS-CoV-2 Mpro." Is TRMT1 prone to a high frequency of missense polymorphisms? If so, then this point makes sense. If not, it is not clear if this really informs on any biologically relevant mechanism.

Given (i) that primate TRMT1 was previously identified under positive selection (i.e. rapid evolution) in an evolutionary screen (Cariou et al PNAS 2022) and (ii) that our study is mostly in vitro, we thought it was important to, first, make sure that this sequence of TRMT1 used in functional assays is not specific to a reference sequence that we tested in vitro, but is actually the sequence of TRMT1 in the human population. Further, we were also looking for whether some variations in the Mpro cleavage site of TRMT1 were possibly present in some humans (could these be linked with severe COVID or susceptibility, for example?).

Overall, this statement aims to anchor our in vitro results to the TRMT1 sequences actually present in humans. However, we agree this does not inform “biologically relevant mechanism”. We therefore took out the “Importantly” that was probably misleading.

"TRMT1 engages the Mpro active site in a distinct binding conformation."This is reported as an observation with little analysis. What is the structural basis of this conformational difference between the bound peptides? Why are the psi angles different? Is there a steric factor that is different between these peptide chains? This section can be substantially improved in detail from its current state.

See our related answer to the next comment below.

"Molecular dynamics simulations suggest kinetic discrimination happens during later steps of Mpro-catalyzed substrate cleavage." This section could have partly addressed my previous comment. It is not clear why there is such a large difference in the psi-angle. With access to several peptide-bound structures, the authors should derive and provide insights into the underlying fundamental principles. After all, this is a major point of discovery in their investigation.

We agree that it is not entirely clear why TRMT1 seems to favor the P3’-in conformation when binding to Mpro. The only other known peptide-bound structure that adopts a similar P2’ psi angle is nsp6/7, but there are not clear sequence, steric, or interaction features that distinguish TRMT1 and nsp6/7 from the other 6 peptide-Mpro structures that favor a P3’-out conformation with larger P2’ psi angle. In particular, the identity of the P1’ and P3’ residues, which would probably be expected to have the largest impact on this conformation, have no clear commonality in TRMT1 and nsp6/7 that give hints about why these adopt this unique conformation. As we describe in the discussion section of the manuscript, and has been observed by many other studies of Mpro, the protease active site is very plastic and able to accommodate a diverse range of sequences surrounding the invariant P1 Gln. Furthermore, while the crystal structures of TRMT1 and other nsp cleavage sequences bound to Mpro show a single peptide conformation in the active site, our MD simulations suggest that both P3’-in and P3’-out type conformations are present in solution for TRMT1, nsp4/5, and nsp8/9, just with different populations. It is very likely that there is a delicate energetic balance between these conformations that may depend subtly on multiple sequence features of the peptide and how they interact with each other and the flexible Mpro active site. As with our replies to questions from Reviewer 2 above about deciphering the underlying principles that connect peptide sequence to cleavage efficiency, we expect that dissecting the detailed links between sequence and binding conformation will be a long-term challenge for mechanistic and biocomputational groups focused on viral protease enzymes; systematic mutation of all residues in the cleavage sequence to multiple different amino acid identities followed by structure determination either experimentally and/or computationally will likely be required to uncover the key sequence or steric properties and interactions that underly and drive favored peptide binding conformations.

To highlight these questions as significant and difficult future challenges toward understanding the fundamental principles underlying SARS-CoV Mpro proteolysis, we have added an additional paragraph (second from the last paragraph) in the discussion section.

This work can be taken to a whole new level if the authors were to provide insights into how TRMT1 degradation by Mpro affects host cell biology and how the inhibition of this activity affects CoV biology.

We certainly agree that showing the biological effects of TRMT1 degradation on host cell biology and/or viral biology could raise the impact of this work. But as discussed in more detail above in our response to the weakness listed in Reviewer 3’s public review, we see the main focus of this work as showing the biochemical and structural basis for TRMT1 recognition and cleavage by SARS-CoV-2 Mpro, and directly showing the immediate effects of this cleavage on the TRMT1-tRNA interaction and modification activity. As was the case with other viral proteases, like the HIV-1 protease, understanding the potentially diverse and nuanced downstream biological effects of host protein cleavage and its impacts on cellular phenotypes or viral fitness could take many years of careful cell biology and virology work. We hope that our paper provides the key first steps to viral biology labs taking on this significant but important challenge for TRMT1!